# Chromatin dysregulation and DNA methylation at transcription start sites associated with transcriptional repression in cancers

Mizuo Ando [1,2], Yuki Saito[1,2], Guorong Xu[3], Nam Q. Bui [4,5], Kate Medetgul-Ernar[1], Minya Pu[1], Kathleen Fisch [3], Shuling Ren[1], Akihiro Sakai[1], Takahito Fukusumi[1], Chao Liu[1], Sunny Haft[1], John Pang[1], Adam Mark[3], Daria A. Gaykalova[6], Theresa Guo[6], Alexander V. Favorov [7,8], Srinivasan Yegnasubramanian[7], Elana J. Fertig[7], Patrick Ha [9], Pablo Tamayo [1], Tatsuya Yamasoba [2], Trey Ideker[4], Karen Messer[1] & Joseph A. Califano[1,10]

Although promoter-associated CpG islands have been established as targets of DNA methylation changes in cancer, previous studies suggest that epigenetic dysregulation outside the promoter region may be more closely associated with transcriptional changes. Here we examine DNA methylation, chromatin marks, and transcriptional alterations to define the relationship between transcriptional modulation and spatial changes in chromatin structure. Using human papillomavirus-related oropharyngeal carcinoma as a model, we show aberrant enrichment of repressive H3K9me3 at the transcriptional start site (TSS) with methylation-associated, tumor-specific gene silencing. Further analysis identifies a hypermethylated subtype which shows a functional convergence on MYC targets and association with *CREBBP/EP300* mutation. The tumor-specific shift to transcriptional repression associated with DNA methylation at TSSs was confirmed in multiple tumor types. Our data may show a common underlying epigenetic dysregulation in cancer associated with broad enrichment of repressive chromatin marks and aberrant DNA hypermethylation at TSSs in combination with MYC network activation.

[1] Moores Cancer Center, University of California San Diego, 3855 Health Sciences Dr, La Jolla, CA 92093, USA. [2] Department of Otolaryngology - Head and Neck Surgery, Graduate School of Medicine, University of Tokyo, 7-3-1 Hongo, Bunkyo-ku, Tokyo 113-8655, Japan. [3] Department of Medicine, Center for Computational Biology and Bioinformatics, University of California San Diego, 9500 Gilman Drive, La Jolla, CA 92093, USA. [4] Department of Medicine, University of California San Diego, 9500 Gilman Drive, La Jolla, CA 92093, USA. [5] Department of Medicine (Oncology), Stanford University School of Medicine, 875 Blake Wilbur Dr, Palo Alto, CA 94304, USA. [6] Department of Otolaryngology - Head and Neck Surgery, Johns Hopkins University School of Medicine, 601 N Caroline St, Baltimore, MD 21287, USA. [7] Department of Oncology, Sidney Kimmel Comprehensive Cancer Center, Johns Hopkins University School of Medicine, 401 N Broadway, Baltimore, MD 21231, USA. [8] Laboratory of Systems Biology and Computational Genetics, Vavilov Institute of General Genetics, Russian Academy of Sciences, Gubkina str. 3, Moscow 119333, Russia. [9] Department of Otolaryngology - Head and Neck Surgery, University of California San Francisco, 2380 Sutter St, San Francisco, CA 94115, USA. [10] Division of Otolaryngology - Head and Neck Surgery, Department of Surgery, University of California San Diego, 9300 Campus Point Drive, La Jolla, CA 92037, USA. Correspondence and requests for materials should be addressed to J.A.C. (email: jcalifano@ucsd.edu)

Epigenetic abnormalities are heritable and collaborate with genetic changes to cause the evolution of cancer. Global hypomethylation at non-coding regions and focal hyper-methylation, typically at promoter-associated CpG islands associated with gene silencing, are hallmarks of cancer cells and this process may be linked to a gain of histone repressive marks[1–3]. Epigenetic regulatory mechanisms modulate cell-specific transcription by packaging DNA into chromatin; combining these epigenetic profiles with gene expression data have provided significant insights into the biological processes such as development, differentiation, and proliferation[4,5]. In addition, the potential reversibility of these epigenetic alterations makes them attractive targets for therapeutic intervention.

Although DNA methylation at promoter-associated CpG islands is well known to be correlated with gene repression[1], genome-wide studies utilizing massively parallel sequencing have described a role for DNA methylation outside these well-described CpG sites and thus expanded the focus of epigenetically mediated transcriptional regulation to much larger regions of interest. For example, CpG island shores which extend up to 2 kb from an island were identified as crucial elements for gene regulation with variable DNA methylation level between normal and cancer cells[6], and sites of alternative transcription and enhancer regions including super-enhancers may be targeted by methylation alterations[7,8]. DNA methylation-related transcriptional changes in cancer cells are often consistent with increasing density of methylation, but whether density itself or spreading toward specific regions is correlated with gene silencing is not clear[9]. The discovery of clinically relevant methylation analysis would preferably be broader and CpG density-independent.

Broad, genome-wide analyses of CpG methylation and chromatin structure in solid tumors have been challenging as they are more resource and computationally intensive than broadly available whole genomic sequencing approaches and ChIP-seq experiments have been challenging in primary tumors and have focused on cell line systems. Large-scale data sets, i.e. the Cancer Genome Atlas (TCGA), have employed array-based analyses of DNA methylation that provide coverage limited in density and with limited coverage in areas outside regions immediately proximate to actively transcribed genes, limiting the capacity to examine tumor-specific methylation alterations outside of promoter-associated CpG islands. Whole genome methylation analyses that have examined broad, non-coding regions in intergenic regions have identified large regions (up to several Mb) that are broadly hypomethylated in cancer as well as small regions where loss or shift of methylation boundaries have particular relevance[10,11]. Differentially methylated regions have been reported as common in high-CpG-density promoters and most cancer-related DNA methylation changes correspond to DNA methylation variation among normal tissues, particularly at genes associated with development[6,8]. To gain insight into DNA methylation-related transcriptional alterations in cancers we analyzed data sets that employed a relatively unbiased sequencing-based analysis of chromatin structure, DNA methylation, and transcription in human papillomavirus-related oropharyngeal squamous cell carcinoma (HPVOPSCC) as well as other tumor types using published data sets.

Unlike traditional head and neck squamous cell carcinoma (HNSCC), the major risk factors for HPVOPSCC are not tobacco or alcohol use, and less common somatic mutations in key cancer genes implies that epigenetic mechanisms might drive oncogenesis[12,13]. Viral oncoproteins E6 and E7 of HPV16 are known to modulate the DNA methylation; E6 can induce upregulation of the DNA methyltransferase *DNMT1* via suppression of p53, whereas E7 can directly bind to and activate DNMT1[14]. Indeed, recent studies using probe-based methods reported that HPVOPSCC have higher levels of gene promoter methylation compared with HPV-negative HNSCC[15]. Here we describe a tumor-specific shift in association of transcriptional repression from DNA methylation at conventional promoter regions in normal tissue, to cancer-specific transcriptional repression associated with broad repressive chromatin marks and DNA methylation at transcriptional start sites (TSSs) regardless of CpG island presence. These findings are complemented by identification of a hypermethylation phenotype in HPVOPSCC, characterized by MYC pathway activation and mutation in chromatin-regulating gene *CREBBP/EP300*.

## Results

**Overview of DNA methylation in HPVOPSCC.** To determine functionally relevant methylation events in HPVOPSCC, we performed MBD-seq and RNA-seq analysis on 47 primary HPVOPSCC tumors and 25 normal oropharyngeal mucosal tissues. Clinical characteristics are previously described[16]. Utilizing MACS-processed MBD-seq data we divided the entire human genome into 100 bp regions and scored each region with a binary methylation value. We then compared DNA methylation level between tumor and normal tissues, demonstrating that tumor-associated alterations in DNA methylation profile occurred on a genome-wide scale (Supplementary Figure 1).

To explore the general effect of DNA methylation on gene expression in relation to TSS location and CpG island presence, for each gene we plotted the methylation levels at each 100 bp segment in a region surrounding the gene's TSS ± 5 kb according to gene expression divided into quartiles. Comparing tumor and normal data sets across all genes and samples; we analyzed genes with CpG island(s) in the promoter region (referred to as 'CGI genes') and those without (referred to as 'noCGI genes') separately. Within these categories, we further divided genes by their expression level from low (Q1- first quartile) to high (Q4- fourth quartile). We then plotted both the individual and mean methylation ratios for all genes within each quartile, by distance from the TSS. To separate out the effects of CpG methylation on large stretches of DNA with variable CpG content and location, we analyzed genomic regions within CpG islands surrounding the TSS ± 5 kb separately, and found a reduction in DNA methylation levels proximal to the TSS (approximately ± 1 kb) for both normal and tumor tissues in actively expressed CGI genes (Fig. 1a). CGI genes with low expression levels showed higher DNA methylation levels around the TSS in tumor compared to normal tissues, implying DNA methylation-mediated silencing of tumor suppressor genes is more pronounced in transcriptionally silent genes with promoter-associated CpG islands. Interestingly, the increase in methylation associated with cancer compared to normal samples in transcriptionally repressed CGI genes extended to include CpG islands located several kb both 5' and 3' from TSSs. DNA methylation levels were also examined in regions excluding CpG islands surrounding the TSS ± 5 kb, and transcriptionally repressed CGI genes showed higher methylation levels in cancer compared to normal, yet similar reductions of methylation around TSS for actively transcribed genes were observed (Fig. 1b), suggesting the importance of DNA methylation and transcription at regions with lower CpG density. In addition, transcriptionally repressed genes not associated with CpG islands promoters (noCGI genes) demonstrated high methylation levels centered around the TSS in both normal and tumor samples (Supplementary Figure 2).

**TSS DNA methylation and transcription in HPVOPSCC.** Next we sought to profile the distribution of genomic regions that are

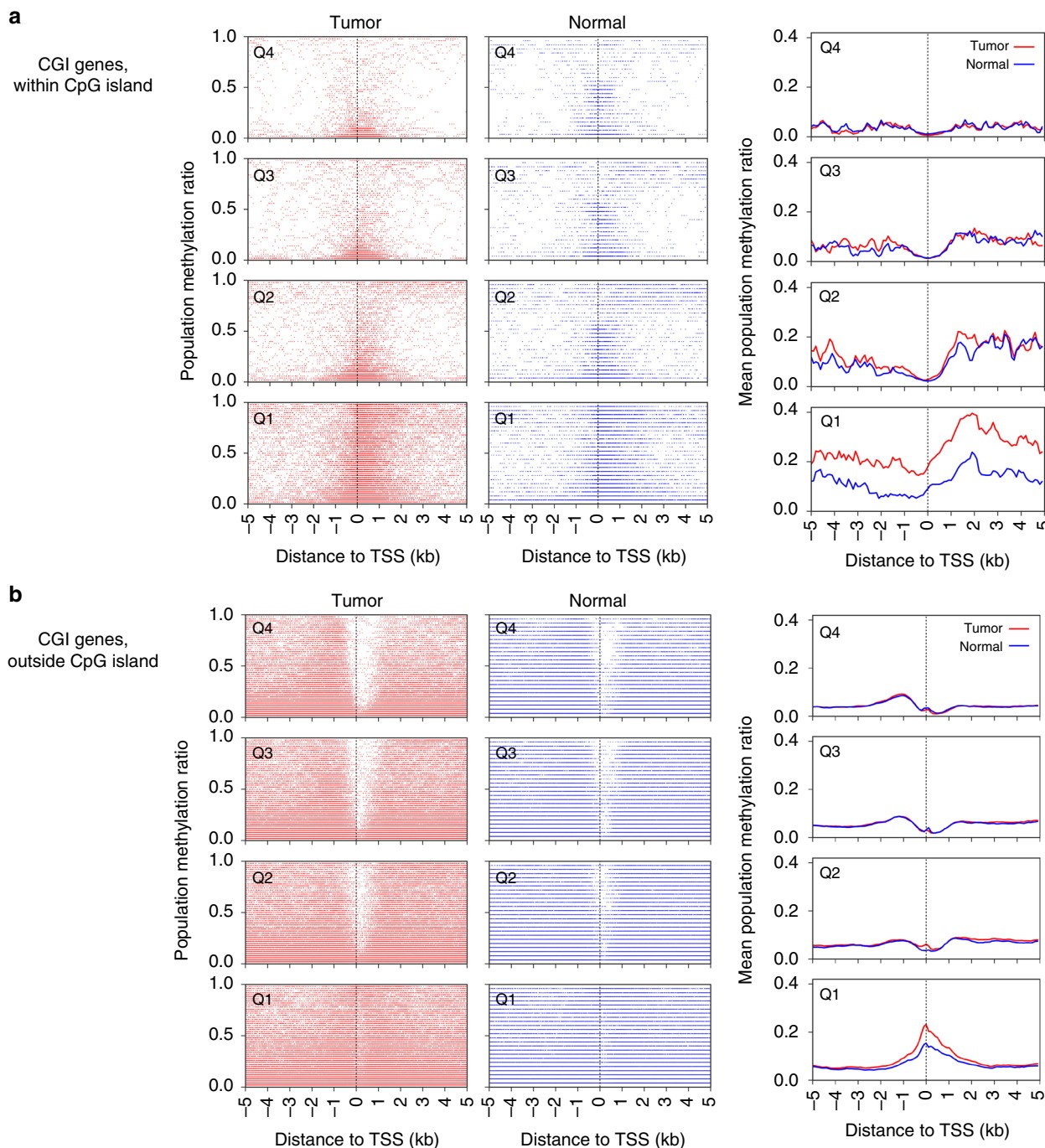

**Fig. 1** DNA methylation and gene expression in HPVOPSCC and normal oropharynx. Scatter plots (left panels) separated by gene expression quartiles based on the expression levels in either tumor or normal samples ($n = 3290, 3289, 3289, 3290$ for Q1, Q2, Q3, Q4, respectively; Q4 is the highest expression) showing methylation ratio at 100 bp segments including genomic loci within (**a**) and outside (**b**) CpG islands in genes with promoter-associated CpG islands (CGI genes, $n = 13158$). CGI genes have the CpG island-containing promoter by definition, though they may have additional CpG island(s) outside of the promoter region. **a** Shows methylation status at each 100 bp segment with such CpG island. As CGI genes also have various methylation profiles at CpG sites outside of the island(s), i.e. shore(s) etc., such 100 bp segments are plotted in **b**. Thus non-overlapping 100 bp segments of an identical set of CGI genes are shown in **a**, **b**. Right panels show the mean values. TSS transcription start site

potentially involved in aberrant epigenetic transcriptional regulation.

To infer which gene-specific regions are specifically involved in gene activation in normal and tumor states, we identified the genomic regions most closely associated with transcriptional activation and repression for all expressed genes. We measured the correlation of gene expression with DNA methylation across TSS ± 5 kb region for tumor and normal samples separately. We then plotted the distribution of significant genomic coordinates spatially in relation to the TSS. In normal tissue, methylation status 1 kb upstream of TSS tended to be more highly associated with expression of genes, consistent with traditional models of

modulation of expression in association with promoter methylation status. Unexpectedly in tumors, we found that the genomic region where DNA methylation was most significantly correlated with gene expression was located directly at the TSS (Fig. 2a). This shift was particularly striking in those genes that contained CpG island(s) within the promoter region (CGI genes) and had a negative methylation-expression correlation. We confirmed these findings by a permutation test comparing the areas under the curve (AUCs) between ± 500 bp from TSS for tumor and normal in each density plot (Fig. 2b). Two representative genes with negative correlation between DNA methylation and RNA expression level are shown in Fig. 2c. *ALDH7A1* was generally hypomethylated in normal tissues and actively expressed, whereas hypermethylated and repressed in one third of tumor samples. In contrast, *SEPT10* was generally hypomethylated and active in tumors.

To validate the MBD-seq data, we also examined differential DNA methylation status between tumor and normal samples in two representative genes using bisulfite sequencing of primary tumor samples (Fig. 2d). We confirmed that tumor-specific methylation changes are not always mere methylation gain in CpG island around TSSs, but instead accompanied by loss of methylation distant from the TSS. These findings implied that methylation events at low CpG density regions are also related to cancer-related gene expression alterations.

**DNA methylation subtypes within HPVOPSCC.** Seeking insights into the DNA methylation-based heterogeneity of HPVOPSCC, we selected 59 genes that exhibited strongly significant (FDR $q < 0.001$) negative correlation between methylation levels within the TSS ± 5 kb region and RNA expression levels in our discovery cohort (Supplementary Data 1). Among selected 59 genes, 52 (88%) showed increased methylation and were repressed in tumor compared to normal samples, suggesting tumor-suppressive function of these dysregulated genes (Fig. 3a). We then conducted unsupervised hierarchical clustering using this gene set and identified three methylation subtypes in our discovery cohort (Fig. 3b). Validation of this gene set in a separate TCGA HPVOPSCC cohort yielded similar results (Fig. 3c), demonstrating that integrated DNA methylation and gene expression analysis could define methylation subtypes in this tumor type.

To explore which methylation subtype drives the changes of aberrant DNA hypermethylation across TSSs between tumor and normal samples in Fig. 2a, we plotted the distribution of significant genomic coordinates in high, intermediate, and low methylation subtypes separately and found the hypermethylation phenotype with negative methylation-expression correlation as the dominant subgroup contributing to this tumor-specific shift to TSSs (Supplementary Figure 3).

**Repressive histone marks around TSS in HPVOPSCC.** Using cancer-unaffected normal mucosa and tumor PDX models derived from highly and lowly methylated tumors defined in our discovery cohort, we performed genome-wide analysis of chromatin and DNA methylation status (Fig. 4a), demonstrating that DNA hypermethylation occurs at the TSS specifically in the highly methylated tumor. Supplementary Figure 4a shows parallel DNA methylation status of PDX models and the parental tumors for representative genes in Fig. 3a. We have also confirmed that the PDX models were similar to the parental tumors by comparing the RNA-seq gene expression profiles[17]. Interestingly, *ALDH7A1*, a representative gene with negative methylation-expression correlation (hypermethylated in tumor), showed a gain of repressive H3K9me3 at TSS and a shift of sharp DNA

methylation peaks toward TSS, particularly in the highly methylated tumor (Fig. 4b). *SEPT10*, another representative gene with negative correlation (hypomethylated in tumor), showed no obvious gain of H3K9me3 and loss of DNA methylation at the promoter-associated CpG island in tumors, with an enrichment of active H3K4me3 around the TSS.

We next compared ChIP-seq data for active H3K4me3 and repressive H3K9me3 histone marks using normal oropharyngeal tissue as well as these PDX models generated from highly and lowly methylated tumors present in our discovery data set. When mean read counts were plotted, H3K4me3 was found to be enriched across the region proximal to the TSS annotated for actively expressed genes (Fig. 4c). The TSS was generally devoid of nucleosomes and stronger enrichment of H3K4me3 occurred 1 kb downstream. No obvious difference between normal and tumor samples was found, though the latter showed broader range of mean signal levels. In contrast, repressive H3K9me3 in normal samples was underrepresented focally at TSS but broadly, and densely distributed in tumors, demonstrating slightly higher signal levels in inactive genes and a higher signal in the high-methylation tumor. Interestingly, for tumor samples, we found uniformly high levels of enrichment of H3K9me3 throughout TSS ± 5 kb regions in inactive genes, whereas biphasic peaks occurred within TSS ± 1 kb in active genes. Overall, the GREAT algorithm revealed that tumor-specific H3K4me3 peaks were enriched for oncogenic signaling including p53, EGFR, differentiation and cell adhesion pathways, whereas normal-specific H3K4me3 showed immune response pathway signatures (Supplementary Figure 4b). Since H3K9me3 depletion at TSS was not seen in tumors compared to broad depletion over several kb seen in normal samples, especially in inactive genes, this implies that there is broad dysregulation of repressive histone marks in tumor across inactive and actively transcribed genes.

We then focused on 407 genes with significant (FDR $q < 0.05$) methylation-expression association between DNA methylation and gene expression in our discovery cohort. When plotted against 1000 randomly selected control gene sets of the same size, we found that H3K9me3 enrichment for these genes was constantly higher throughout TSS ± 5 kb regions in both tumor and normal samples, in contrast to H3K4me3 within the range of background levels (Fig. 4d). Observed H3K9me3 profiles in each tumor sample were similar to those of inactive genes in Fig. 4c, in line with the fact that most of these 407 genes had negative methylation-expression correlation and were silenced. Of note, H3K9me3 depletion at TSS was less evident in highly methylated tumor samples, indicating that changes in repressive H3K9me3 marks might be related to DNA methylation levels.

**TSS DNA methylation and transcription in TCGA HPVOPSCC.** To validate these findings and exclude potential methylation platform bias from our initial cohort analyzed using MBD-seq, we analyzed 54 HPVOPSCC and 20 normal samples available from TCGA cohort that uses a probe-based Infinium HumanMethylation 450 K BeadChip (HM450K) microarray. Of note, probes upstream of a gene are mainly located within −1.5 kb of the TSS in HM450K platform, DNA methylation data beyond this region was scarce, and provides a significant limitation on interrogating genomic regions more than 1.5 kb upstream of TSS. First we plotted DNA methylation levels at each HM450K probe in TSS ± 5 kb region according to the quartiles of gene expression for CGI genes. As expected, we found low DNA methylation levels of probes within CpG islands around TSS annotated for actively expressed CGI genes, and tumors exhibited higher levels of DNA methylation for transcriptionally repressed CGI genes over a large area extending up to the limit of probe

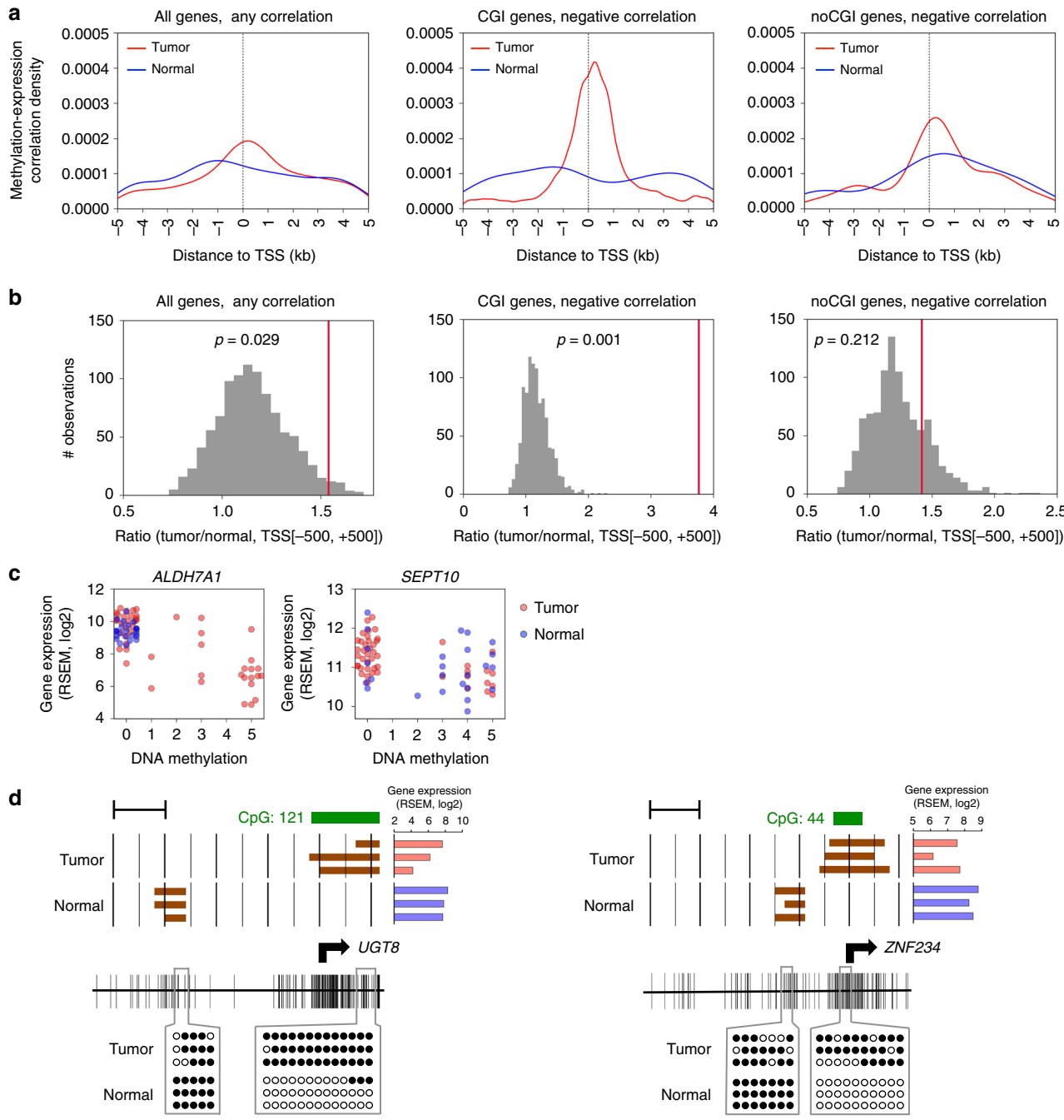

**Fig. 2** Association of DNA methylation at TSS with gene expression in HPVOPSCC. **a** Methylation-expression correlation density plots showing where DNA methylation and gene expression are significantly correlated (FDR q < 0.05), demonstrating a strong, tumor-specific association of expression with methylation at the TSS. Y-axis represents probability and the area under the curve (AUC) adds up to 1. Plots of any correlation in all genes (n = 19866) in our discovery HPVOPSCC data set, and negative correlation in genes with and without promoter-associated CpG island (CGI genes (n = 13158) and noCGI genes (n = 6708), respectively) are shown. **b** Significance of DNA methylation expression associations in tumor. The red line represents the observed AUC ratio (tumor/normal) at TSS ± 500 bp region. Histogram represents the null distribution as calculated by multiple permutations (n = 1000). Results of any correlation in all genes, and negative correlation in CGI genes and noCGI genes are shown. **c** Scatter plots of two representative genes for expression and DNA methylation levels at the most significantly correlated 500 bp window determined by MBD-seq. **d** Bisulfite sequencing confirmation of MBD-seq data from each two regions of two representative genes. The arrow indicates the TSS of each gene. Brown bars in the upper panel indicate binary methylation value at each 100 bp segment determined by MBD-seq. Red and blue bars show mRNA level determined by RNA-seq. Vertical lines in the lower panel represent CpG sites. Open and filled circles denote unmethylated and methylated CpG sites determined by bisulfite sequencing, respectively. Scale bars, 1 kb

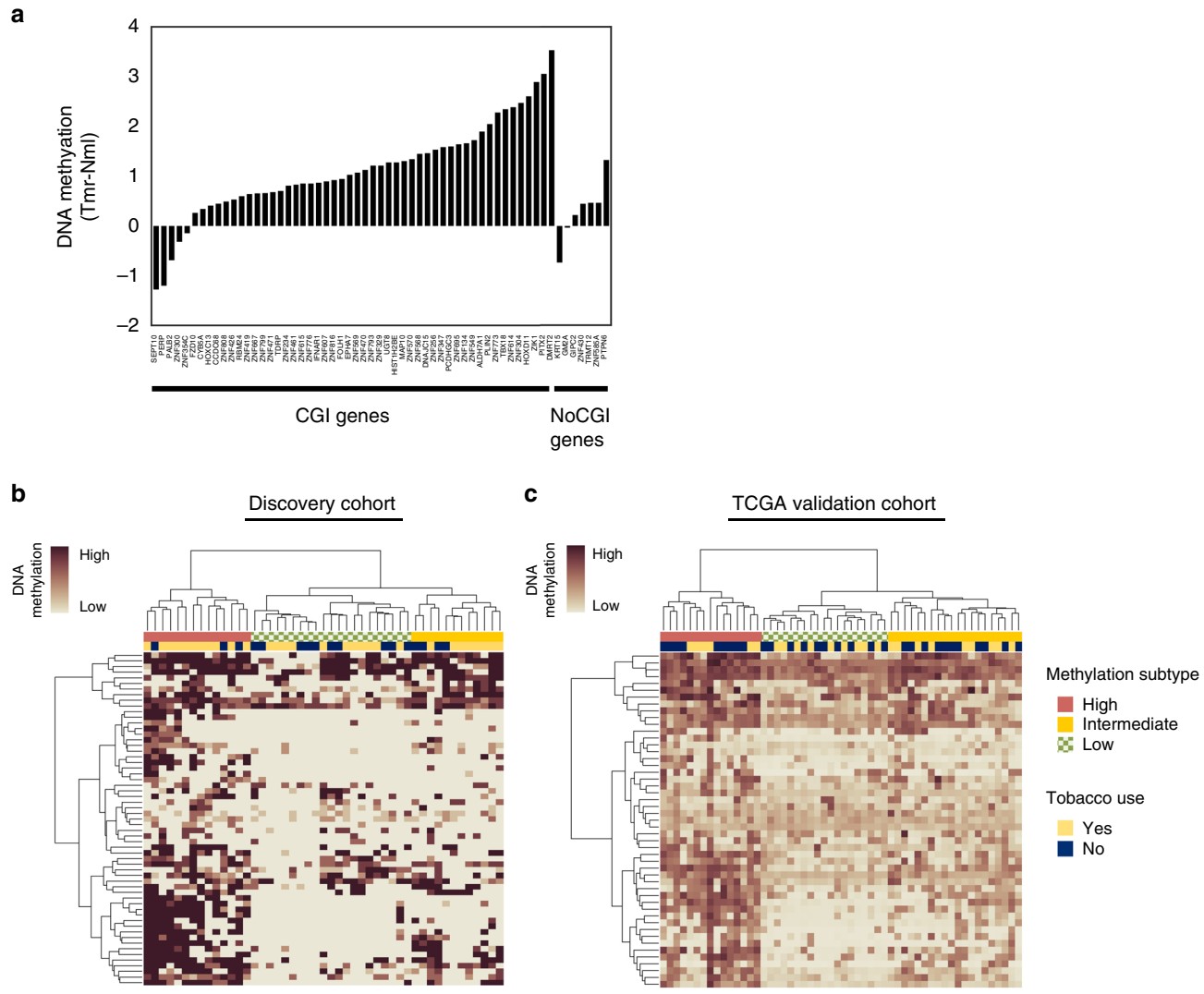

**Fig. 3** Unsupervised hierarchical clustering based on DNA methylation-expression correlations. **a** Changes in DNA methylation levels of 59 genes for the discovery cohort. **b, c** Clustered heatmaps for methylation levels of the genes (rows) and HPVOPSCC tumor samples (columns). Identical gene set was used for the discovery cohort (**b**, $n = 47$) and the TCGA validation cohort (**c**, $n = 54$)

presence (1.5 kb) upstream and for several kb downstream (Fig. 5a). For regions excluding CpG islands, transcriptionally repressed CGI genes showed slightly higher methylation levels in cancer compared to normal, and decreased methylation around TSSs were observed for active genes (Fig. 5b). These observations validated our findings in the MBD-seq data in our initial cohort.

We then applied a correction to the probe design bias in the HM450K platform used in TCGA data sets by computing a weighted mean of methylation levels across TSS ± 5 kb regions for each gene, and profiled the distribution of probes with significant methylation-expression correlations. A similar increase of a density peak at TSS in tumor samples was observed, without an obvious shift toward TSS, probably due to insufficient probes in upstream region beyond −1.5 kb (Fig. 5c). However, we noted a tumor-specific significant increase in negative methylation-expression correlation at the TSS in CGI genes that was confirmed by permutation testing (Fig. 5d).

**TSS DNA methylation and transcription in multiple cancers.** We extended our findings to determine whether the association of TSS DNA methylation association with transcription was unique

to HPVOPSCC or if it could also be broadened to other tumor types. We analyzed HPV-negative head and neck (HNSC), colon (COAD), and breast (BRCA) cancers with corresponding normal tissue available from TCGA. Similar to findings in HPVOPSCC, we found that a tumor-specific significant increase in negative methylation-expression correlation at the TSS in CGI genes in HPV-negative HNSC and breast cancer. A similar relationship was noted in colon adenocarcinoma, albeit less pronounced (Fig. 5e–g). Finally, we tested salivary adenoid cystic carcinoma (ACC) using an analogous MBD-seq and RNA-seq data sets available from our cohort of previous study[18]. Utilizing 15 tumor and 14 normal samples with an identical computing pipeline to the current study, we did not find a shift in distribution of genomic regions having methylation-expression correlations despite the fact that the promoter region tended to be more highly associated in normal tissues as expected (Supplementary Figure 5). Although difference in overall methylation status between tumor types as well as the batch effect issue should be taken into consideration, these results demonstrate that a cancer-specific association of aberrant transcriptional silencing is focused on the TSS, rather than CpG island promoters in multiple, but not all tumor types.

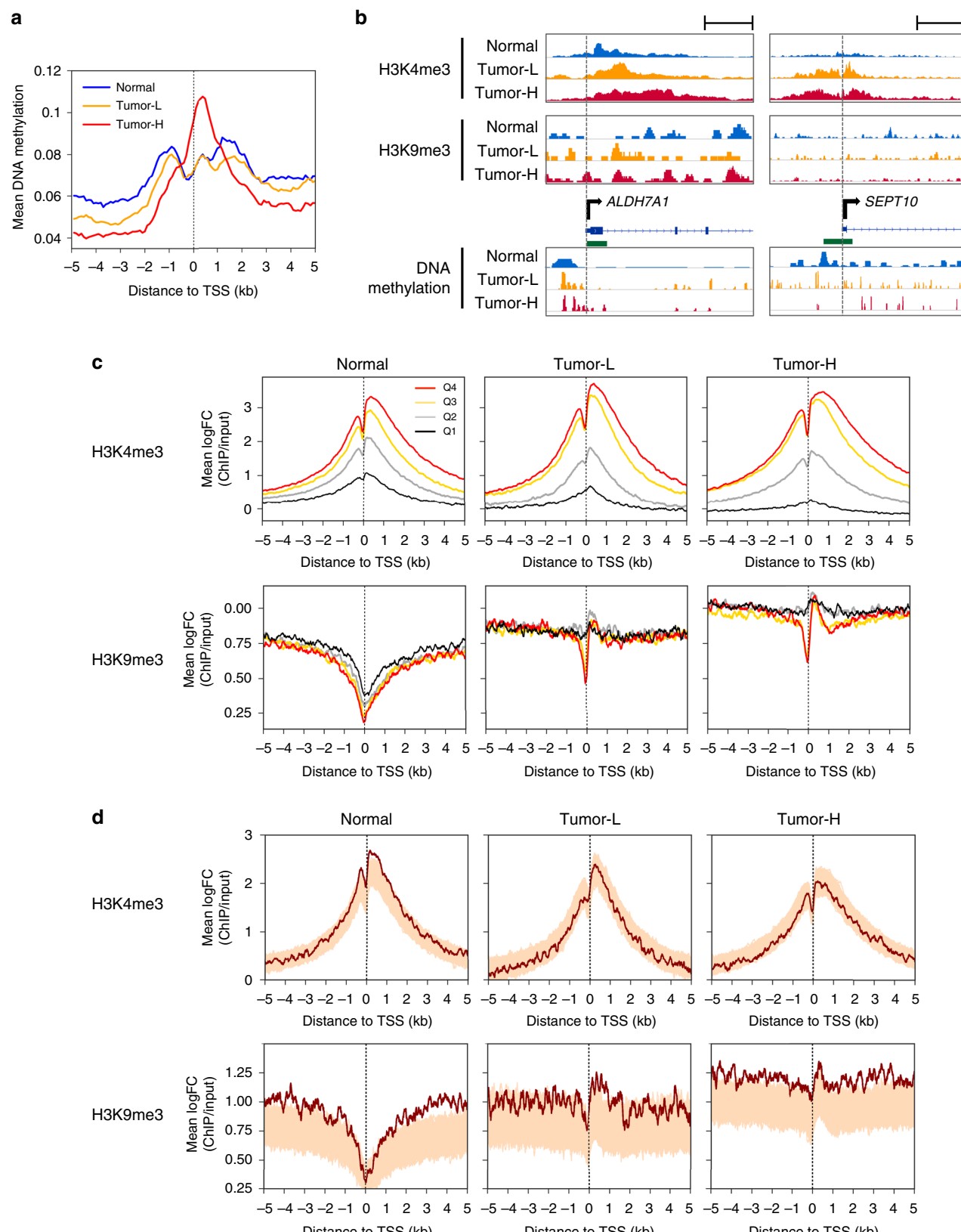

**MYC pathway associated with highly methylated HPVOPSCC.** In order to identify key networks associated with DNA methylation subtypes, we identified differentially expressed genes in the highly methylated tumors compared to lowly methylated tumors and performed ssGSEA in our original HPVOPSCC cohort. Interestingly, ssGSEA using motif gene sets revealed that the transcriptome of the highly methylated subtype is enriched for motifs related to MYC (Fig. 6a). MYC dimerizes with its partner

**Fig. 4** Chromatin structure correlation in normal oropharynx and PDXs. **a** DNA methylation profiles of the normal oropharyngeal tissue and highly/lowly methylated tumors. Mean DNA methylation value of RefSeq genes ($n = 20013$) determined by MBD-seq are shown. **b** ChIP-seq data for H3K4me3 and H3K9me3 histone marks and MBD-seq data of two representative genes (see also Fig. 2c). Genes (blue) and CpG islands (green) are presented in a forward fashion. Scale bars, 1 kb (left) and 5 kb (right). **c** Mean H3K4me3 and H3K9me3 levels of all genes in 10 bp windows separated by gene expression quartiles (Q4 is the highest expression). Plots for normal oropharyngeal tissue, lowly (Tumor-L), and highly (Tumor-H) methylated tumors are shown. The number of genes for Q1, Q2, Q3, Q4; $n = 4366, 4373, 4358, 4382$ for Normal, $n = 4366, 4336, 4407, 4370$ for Tumor-L, and $n = 4370, 4369, 4370, 4370$ for Tumor-H, respectively. **d** Mean H3K4me3 and H3K9me3 levels of 407 genes with significant correlation between DNA methylation and gene expression in our discovery cohort. The shaded region shows signal of 1000 randomized gene lists of the same size ($n = 407$) selected from the RefSeq genes, thus represents the genomic background for each window

MAX and directly binds to a consensus DNA sequences, known as Enhancer box (E-box)[19]. We then expanded three gene sets to show genes leading to the enrichment of these MYC-related motifs (Supplementary Data 2). It is noteworthy that DNA methyltransferase 3 alpha (*DNMT3A*) was unexpectedly enriched in the lowly methylated subtype. ssGSEA using hallmark (Fig. 6a, Supplementary Data 3) and curated gene sets (Supplementary Data 4) were also applied, revealing that pathways related to MYC are enriched in a concordant way though there was no significant difference in MYC expression itself (Fig. 6b).

Interestingly, inactivation of chromatin modifier p300, a homolog of the histone acetyltransferase CREBBP, has been associated with deregulation of MYC transcription in lung cancer cell lines, and loss of MYC expression in the context of *CREBBP* and *EP300* loss is synthetically lethal[20,21]. To determine if this were a component of the hypermethylation phenotype associated with MYC pathway alteration in HPVOPSCC, we performed whole-exome sequencing of our initial discovery cohort and found a statistically significant association of *CREBBP/EP300* mutation with the hypermethylation phenotype, further supporting a link between MYC pathway activity and transcriptional dysregulation related to chromatin alterations in this phenotype (Fig. 6c). We found total five *CREBBP* mutations; E1566X in Tumor-L, and Q771X, R1446C, E1550K, and Q2202_Q2203del in Tumor-H, four of which affect the HAT (histone acetyl transferase, location 1342–1649) domain as has been reported for other cancer types[22,23].

We further validated these findings in the TCGA HPVOPSCC cohort. Likewise, ssGSEA analyses based on methylation subtypes determined by unsupervised clustering (Fig. 3c) showed that MYCMAX_01 motif was significantly enriched ($p = 0.00132$, an empirical phenotype-based permutation test procedure), confirming the functional convergence on MYC pathway in highly methylated subtype in this tumor type (Supplementary Figure 6a).

**p300-mediated MYC deregulation alters DNA methylation**. To obtain the mechanistic insight driving the observed genetic and epigenetic changes, we used HPV-positive UM-SCC-47 cell line (*CREBBP* Q1092X mutant, *EP300* wild type) as a model. *CREBBP* Q1092X is a loss-of-function mutation unable to interact with histone, and knockdown of *MYC* suppressed growth of these cells as previously reported in the *CREBBP*-deficient cancer types (Supplementary Figure 6b)[20].

Firstly, we demonstrated that *EP300* knockdown significantly reduced *MYC* mRNA and protein levels leading to inhibition of the proliferation of these cells (Fig. 6d) and confirmed that the *EP300*-mediated MYC suppression was due to reduced histone H3K27ac, which is also consistent with the previous study (Fig. 6e)[20]. We then examined the impact of *EP300* knockdown on *CDK2*, one of the MYC target genes that had significantly different expression levels between high and low methylation subtypes both in our discovery and the TCGA validation cohorts

(Supplementary Figure 6c). As expected, *EP300* knockdown significantly reduced occupancy of MYC at the *CDK2* promoter and resulted in decreased expression level (Fig. 6f). Finally, we evaluated alterations in DNA methylation induced by *EP300* knockdown. We selected three representative genes in Fig. 3a, *ZNF470*, *ZNF568*, and *ZNF569*, whose expression level had a negative correlation with that of *MYC* both in our discovery and the TCGA validation cohorts (Supplementary Figure 6d). We found that DNA methylation status of these genes decreased significantly indicating the dependence on MYC pathway alteration in the *CREBBP*-deficient context (Fig. 6g).

## Discussion

Epigenetic changes have been associated with transcriptional dysregulation in human cancers, although the understanding of the nature of those associations has evolved[1–3,6]. Global hypomethylation at non-coding regions and focal hypermethylation at promoter-associated CpG islands associated with gene silencing, have been seen as the dominant hallmarks of DNA methylation alterations in human cancers. These DNA methylation changes have been accompanied by chromatin structural alterations although the interaction of these histone and DNA methylation alterations in tumors continues to be defined. In this study, we define a coordinated alteration of epigenetic changes at the TSS in tumors that include broad repressive chromatin marks around genes in tumor with focal loss of repression at the TSS for expressed genes, as well as tight association with DNA methylation and high levels of repressive chromatin marks at the TSS for repressed genes.

These findings are consistent with prior observations including the following (1) the negative association of CpG island promoter methylation approximately 1 kb 5′ to the TSS with gene-specific transcriptional activation in normal head and neck mucosa, and (2) higher levels of tumor-specific methylation in broad areas, including CpG shores, islands, and the TSS at CGI genes that have undergone silencing compared to normal tissue. This includes previously noted increases in gene-specific promoter methylation in CpG islands as part of the cancer phenotype, but also indicates that aberrant methylation extends over 5 kb or more and includes broad regions without CpG island presence[1,3,6,24,25].

Additional evidence for a key role in TSS chromatin status in transcriptional dysregulation in cancer comes from ChIP-seq data, that demonstrate normal patterns of H3K4me3 activating marks across tumor and normal samples, however, repressive H3K9me3 marks showed no depletion at tumor TSS in comparison to normals, and higher levels in tumors with a hypermethylation phenotype. This implies that the aberrant DNA methylation and repressive histone marks are related to, and focus on, the TSS for cancer-specific gene alterations, rather than 5-prime CpG island promoters. It has been demonstrated that widespread changes in DNA methylation patterns and chromatin modifications are key features of epigenetic reprogramming in

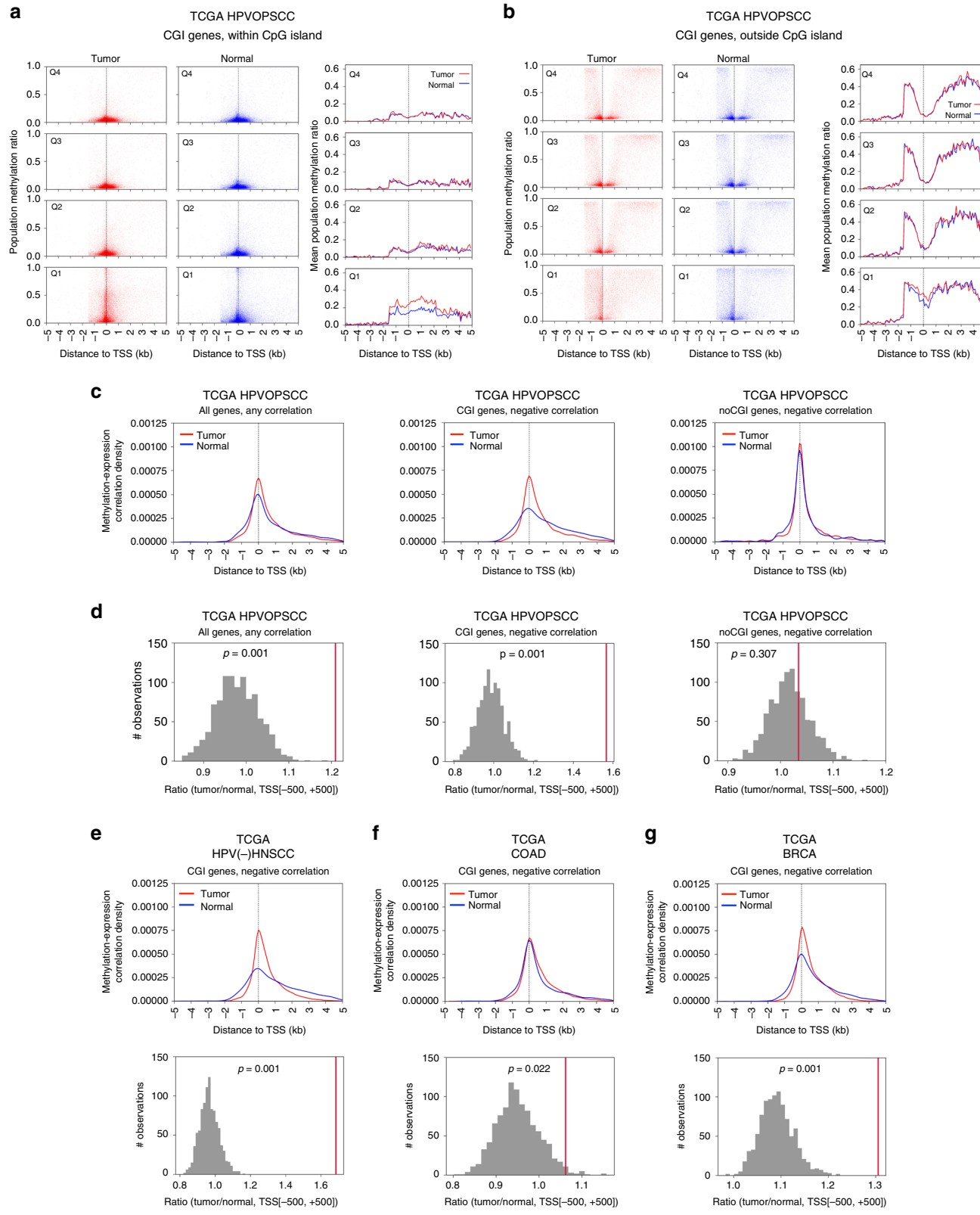

malignant transformation and stem cell differentiation unlike in epithelial-to-mesenchymal transition[1,26,27]. A breast cancer cell line study demonstrated mutually exclusive relationship between DNA methylation and H3K9me3, displaying allelic DNA methylation where one allele is DNA methylated while the other allele is occupied by H3K9 modifications[28]. The mechanism for epigenetic alteration may not be necessarily DNA methylation-specific but could be a loss of structural integrity of euchromatin in TSS that is reflected in methylation changes. It is possible that CGIs are not rigidly positioned around a window around the TSS, and that signals could still partly be derived from CGIs outside of consensus promoter regions. However, we chose the most

**Fig. 5** Validation using TCGA cohort. Note the insufficient data in upstream region beyond −1.5 kb of TSS due to lack of HM450K probes. **a, b** Scatter plots (left panels) separated by gene expression quartiles based on the expression levels in either tumor or normal samples ($n = 3208, 3208, 3208, 3208$ for Q1, Q2, Q3, Q4, respectively; Q4 is the highest expression) showing methylation ratio at HM450K probes located within (**a**) and outside (**b**) CpG islands. Right panels show the mean values. **c** Methylation-expression correlation density plot showing an increase of a density peak where DNA methylation and gene expression are significantly correlated (FDR $q < 0.05$). Plots of any correlation in all genes ($n = 19004$) in the TCGA validation data set, and negative correlation in CGI genes ($n = 12832$) and noCGI genes ($n = 6172$) are shown. **d** Significance of DNA methylation increase in tumor. The red line represents the observed AUC ratio (tumor/normal) at TSS ± 500 bp region. Gray bars represent the null distribution as calculated by multiple permutations ($n = 1000$). Results are also shown for CGI genes ($n = 12873$) among HPV-negative head and neck squamous cell carcinoma (**e**, HPV(-)HNSCC), colon adenocarcinoma (**f**, COAD), and breast invasive carcinoma (**g**, BRCA) available from TCGA

commonly reported and accepted consensus definitions to allow for our results to be compared to prior studies. The implications of these data are indeed that the definition of a promoter region that responds to epigenetic alteration may not universally apply, and that CGI that don't fit current definitions may still be responsive to methylation and histone alteration.

There are additional findings that have not been previously noted. The increased cancer-specific association of transcription with TSS methylation is demonstrated in HPVOPSCC, HPV-negative HNSCC, breast cancer and to a lesser extent in colon cancer. This implies a common change in transcriptional regulation associated with DNA methylation that does not fit the existing paradigm of transcriptional control associated with aberrant methylation of CpG island promoters in these tumor types. Interestingly, the association of gene-specific shore methylation with transcription was previously noted in colon cancer[6], and our examination of colon cancer TCGA data did not show as dramatic a shift to TSS DNA methylation association with transcription in colon cancer compared to other solid tumors, maintaining consistency with the prior study[6]. Although there has been efforts to reveal consistent and ubiquitous methylation changes between tumor and normal tissues on a pan-cancer scale[10,11,29–31], an array-based methylation analysis on normal and tumor pairs of breast, colon, liver, lung, and stomach samples reported that colon cancer was farthest from breast cancer, implying that there appears to be a tissue/tumor-specific epigenetic cancer pathway[32].

Historically, epigenetic alterations in human cancers have shown significant diversity in terms of CpG methylation at gene-specific loci including intragenic regions, CpG shores, enhancer regions, LINE elements, and other classes of regulatory elements[7,24,33,34]. Challenges with determining these relationships includes the following (1) use of cell lines that have been shown to undergo selection pressures that perturb epigenetic profiles in comparison to primary tumors, (2) lack of normal, cancer-unaffected tissue from normal controls for comparison, (3) lack of adequate sample size of primary tumor and normal tissue to drive power for comparison across multiple samples, (4) inherent bias from array-based platforms that preferentially sample loci adjacent to gene promoter regions, and (5) resource limitations associated with whole genomic bisulfite sequencing due to cost that results in limited sample sizes[11,25,32,35]. In this study, we employed sampling of a robust tumor and normal tissue comparison to define the association of chromatin structure and CpG methylation with transcription over a broad region. We used a relatively unbiased whole genome MBD-seq approach to define key local methylation changes associated with cancer-related transcription in a broad 10 kb surrounding the TSS that included CpG shore, CpG island, TSS, and intragenic regions[33]. To demonstrate that these findings were independent of DNA methylation platform, we validated these findings in TCGA data sets that employed a different methylation platform to derive different data sets from multiple solid tumor types. We couldn't

clearly identify DNA methylation subtypes among TCGA HPV-negative HNSCC nor breast cancer when using 59 genes determined by MBD-seq for our discovery HPVOPSCC cohort. We also tried to determine an optimized gene set using each TCGA cohort, but probably due to limited coverage provided by HM450K microarray, subsequently failed to identify DNA methylation subtypes. Although we think that integrated DNA methylation and gene expression analysis might define subtypes in these tumor types, the specific gene set according to tumor type is needed for successful clustering and the discovery analysis would preferably be broader than using HM450K microarray.

To define the biologic pathways underlying this particular phenotype, we used our initial HPVOPSCC cohort for ssGSEA, demonstrating activity of the MYC pathway associated with a hypermethylation phenotype. This was easily validated in the separate TCGA HPVOPSCC cohort, reinforcing the validity of this association. Although previous studies have confirmed a distinct DNA methylation signature in HPV-associated HNSCC when compared to HPV-negative counterpart[15,35–39], few studies including ours demonstrated distinct heterogeneity and possibly causal mechanisms within HPVOPSCC[40–42]. For example, a recent study using DNA methylation-based clustering approach successfully identified an epigenetically distinct subgroup in HPV-negative HNSCCs characterized by H3K36 alteration and DNA hypomethylation, but HPV-positive tumors represented a different subset[39]. Of note, TCGA Network has described a small number of differentially methylated and expressed genes in HPV-associated HNSCC tumors with and without HPV integration though the roles of these genes are yet to be determined[13].

MYC has first been described as a classic transcription factor forming MYC-MAX complexes that binds to E-box[19], but an emerging role for MYC as a global regulator of the cancer epigenome and transcriptome is clear. MYC complexes were shown to recruit DNA methyltransferase 3 alpha (DNMT3A), increasing DNA methylation at specific promoter regions[43]. MYC has also been demonstrated to modify histone marks, preserving active marks such as H3K4 methylation while decreasing repressive H3K9 marks[44]. Another chromatin regulatory protein identified as a direct MYC target is the insulator protein CCCTC-binding factor (CTCF), which is thought to define the boundaries between active and heterochromatic DNA, affecting transcriptional alterations of wide regions of the genome[45]. It is noted that *MYC* is rarely mutated in HNSCC, although these data are consistent with the prior identification of MYC network activation in HNSCC[12,46,47]. Interestingly, the association of *CREBBP* mutation with MYC network activation is concordant with prior data in solid tumors showing synthetic lethality between *CREBBP* mutation and *MYC* inhibition, implying that MYC network activation support carcinogenesis driven by *CREBBP* mutation[20,21]. This also raises the possibility that targeting MYC indirectly in *CREBBP* mutant or other chromatin component mutants could be an attractive therapeutic approach. We were able to demonstrate a phenotype of aberrant transcription

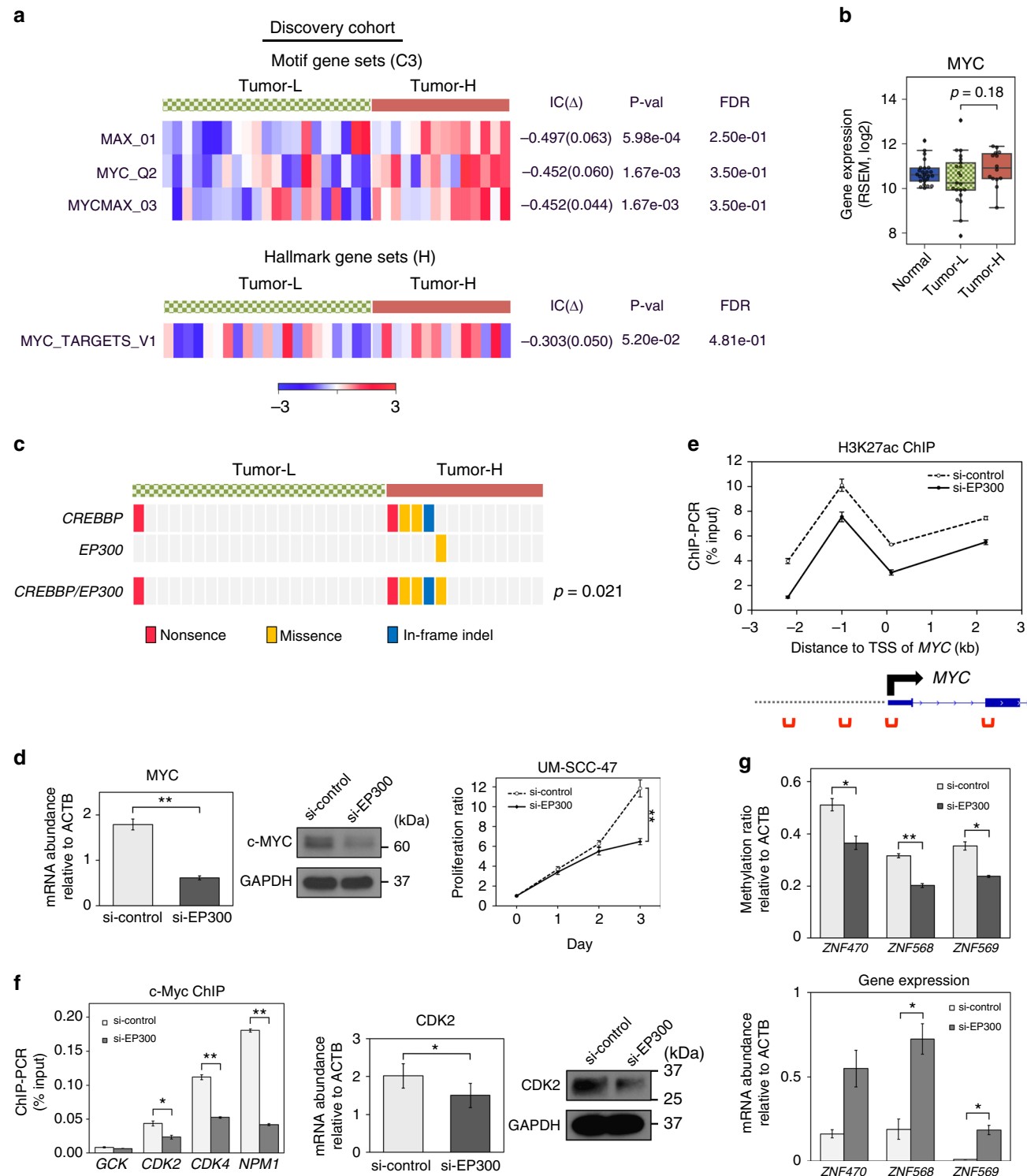

associated with TSS methylation in breast cancer as well as HPV-negative head and neck cancer, but not colon adenocarcinoma. It is possible that the inclusion of MSI (microsatellite instability) colon adenocarcinoma or all colon cancers may not present with a DNA methylation phenotype that demonstrates these associations[30].

Our study had several limitations. First, our discovery cohort consisted of patients with stage 3/4 HPVOPSCC (AJCC/UICC 7th edition) and almost all received postoperative radiotherapy

with or without concurrent chemotherapy. In consequence, only three patients, one from each subtype, experienced disease recurrence. Although the identified subtypes of HPVOPSCC cannot be used for stratification of patients treated with intensive multimodal therapy, we believe that our findings may provide clues to future de-escalation therapy. Second, it is challenging to obtain adequate normal tissues from matched healthy individuals. In this study we analyzed normal oropharyngeal tissues obtained from unmatched controls, as tumor adjacent tissues are known to

**Fig. 6** MYC pathway associated with *CREBBP/EP300* loss in highly methylated HPVOPSCC. **a** Single-sample gene set enrichment analysis (ssGSEA) scores ranked by their degree of association (IC) between highly and lowly methylated HPVOPSCC tumors. MYC-related motif (C3) and hallmark (H) gene sets enriched in highly methylated tumors in our discovery cohort are shown. An empirical phenotype-based permutation test procedure is used to estimate P values. Tumor-H, highly methylated HPVOPSCC. Tumor-L, lowly methylated HPVOPSCC. **b** MYC expression of each sample in our discovery cohort ($n =$ 35, excluding normal samples, t-test). In boxplots, the ends of the boxes and the middle line represent the lower and upper quartiles, and medians, respectively. Whiskers extend to show the rest of the distribution, except for points that are determined to be outliers using a function of the inter-quartile range. **c** *CREBBP/EP300* mutations show significant association with the hypermethylation phenotype Fisher's exact test. **d** Suppression of MYC expression (left and center panel) and growth of UM-SCC-47 cells (right panel) by *EP300* knockdown. mRNA and protein detection were performed on lysates of UM-SCC-47 cells collected at 48 h after siRNA transfection. Growth is normalized to day 0 and measured over 3 days. **e** Reduced histone H3K27ac at *MYC* locus by *EP300* knockdown measured by ChIP-qPCR (upper panel). Red marks in the lower panel indicate the regions ChIP enrichment were measured. The arrow indicates the TSS of *MYC* gene. ChIP assays were performed on UM-SCC-47 cells collected at 48 h after siRNA transfection. **f** Reduced occupancy of MYC at the indicated gene promoter determined via ChIP-qPCR (left panel) and suppression of CDK2 expression (center and right panels) by *EP300* knockdown in UM-SCC-47 cells. *CDK4* and *NPM1* as positive controls[66], and *GCK* as a negative control were used. mRNA and protein detection were performed on cell lysates collected at 48 h after siRNA transfection. **g** Effects of *EP300* knockdown on DNA methylation (left panel) and mRNA expression (right panel) levels in *ZNF470*, *ZNF568*, and *ZNF569*, whose expression level had a significant negative correlation with that of *MYC* in the discovery cohort. All experiments in UM-SCC-47 cells were performed at least in triplicate, and data are expressed as mean ± SE. *P < 0.05 and **P < 0.01, t-test. Source data are provided as a Source Data file

carry genetic and epigenetic alterations. On the other hand, TCGA has a limited number of matched normal tissues and its cohort assembly is biased toward larger, surgically treated late stage tumors. Nonetheless, TCGA is still the largest data set available to date. Finally, although we analytically and experimentally demonstrated the dependence of DNA methylation on MYC pathway alteration in the *CREBBP*-deficient context, further studies with a large, well-characterized clinical sample are warranted to obtain a complete picture of a hypermethylation phenotype in human cancers.

In summary, we show a common underlying mechanism that drives transcriptional dysregulation and silencing in solid tumors that is associated with dense chromatin repressive marks and aberrant hypermethylation at TSSs independent of CpG island presence, and is associated with MYC network activation. The fact that the association of transcription with TSS methylation status is present in multiple, but not every tumor type indicates that there are specific molecular networks that facilitate this pathologic epigenetic program, and that there is potential opportunity to treat cancers with this type of epigenetic dysregulation by targeting this pathway that drive this phenotype.

## Methods

**Patient characteristics**. Primary tumor tissue samples were obtained from a cohort of 47 patients with HPVOPSCC, as previously described[48]. Normal oropharynx tissue from uvulopharyngoplasty (UPPP) surgical specimens were obtained from 25 cancer-unaffected controls. Tissue samples were collected from the Johns Hopkins Tissue Core, part of the Head and Neck Cancer Specialized Program of Research Excellence (HNC-SPORE) under an approved IRB protocol (NA_00036235) and written informed consent was obtained from each patient prior to collection of samples. This protocol also permitted the use of the tumor tissue for patient-derived murine xenograft (PDX) model development, including establishment of two murine PDXs from two separate primary tumors in this cohort. This study qualified for exemption under the U.S. Department of Health and Human Services policy for protection of human subjects [45 CFR 46.101(b)]. Analysis of deidentified data sets at UCSD was performed under an approved IRB protocol. To perform validation within TCGA, publicly available data were obtained for head and neck squamous cell carcinoma (HNSC), colon adenocarcinoma (COAD), and breast invasive carcinoma (BRCA). Primary salivary gland adenoid cystic carcinoma (ACC) tissue and adjacent normal parotid samples were also obtained via the Johns Hopkins Pathology Department under an approved protocol (92-07-21-01)[18].

**RNA sequencing analysis**. RNA was extracted from frozen tissue sections from tissue samples and RNA sequencing was performed. Samples were required to achieve an RNA Integrity Number (RIN) of at least 7.0. In brief, a stranded RNA library was prepared using the TruSeq stranded total RNA seq poly A + Gold kit (Illumina), and then ribosomal RNA reduction was performed and purified with AMPure XP magnetic beads (Beckman Coulter). Sequencing was performed using the HiSeq 2500 platform sequencer and the TruSeq Cluster Kit (Illumina), resulting

in approximately 80 million paired reads per sample. Next, the RNA sequencing data were normalized based on the version 2 protocols developed by TCGA[12]. Alignment was performed using MapSplice2 version 2.0.1.9 to the GRCh37/hg19 genome assembly. Gene expression values were quantified from RNA sequencing data using RSEM version 1.2.9 and upper quartile normalization according to the TCGA RSEM v2 normalization pipeline[12].

**MBD-seq data and analysis**. DNA was extracted from frozen tissue samples and methyl-binding protein domain sequencing (MBD-seq) was performed. Briefly, DNA was sonicated, end-repaired, and ligated to SOLiD P1 and P2 sequencing adaptors lacking 50 phosphate groups (Life Technologies), using the NEBNext DNA Library Prep Set for SOLiD according to the manufacturer's protocol (NEB). Libraries were then nick-translated with Platinum Taq polymerase and divided into two fractions: an enriched methylated fraction that was subjected to isolation and elution of CpG-methylated library fragments by using MBD2-MBD–bound magnetic beads, and a total input fraction that was left unenriched[33]. The resulting libraries were subjected to emulsion PCR, bead enrichment, and sequencing on a SOLiD sequencer to generate on average approximately 25–50 million 50 bp single-end reads per sample according to the manufacturer's protocols. Based on the MBD-seq data, DNA methylation status of each 100 bp segment across the genome was determined with MACS[49] peak calling for each sample (https://github.com/favorov/differential.coverage). Circos version 0.6.9 plotting was used to show the genome-wide difference between tumor and normal in 1 Mb windows[50]. The DNA methylation level in a non-overlapping 500 bp window was determined as the sum of the binary score defined as the presence or absence of a MACS peak for each 100 bp segment. A gene promoter was defined as the region 1,500 bp upstream and 500 bp downstream from TSS. CpG islands data was downloaded from the UCSC genome annotation database for GRCh37/hg19 (https://genome.ucsc.edu). Protein-coding genes were classified into those containing CpG island(s) within the promoter region (CGI genes) and those without (noCGI genes). Gene expression (RSEM normalized) was correlated with DNA methylation level in each 500 bp window using Spearman's rank correlation followed by a false discovery rate (FDR) correction using Benjamini-Hochberg method for multiple comparisons. The FDR adjusted p-values (FDR q-values) below 0.05 were used to define windows with significant methylation-expression correlation. Kernel density estimation was then used for plotting the distribution of significant genomic coordinates as a function of the distance to TSS for the indicated gene group (all/CGI/noCGI) and correlation (any/negative). To obtain a null distribution, DNA methylation and gene expression levels were permuted across the samples 1,000 times and the Spearman's correlation analysis were repeated.

**Bisulfite sequencing**. Primers specific for bisulfite-converted DNA were designed for the region where DNA methylation profile in tumor was noted to be altered both near and distant from TSS on MBD-seq data. Representative highly methylated tumor samples determined by unsupervised clustering of the original cohort were selected. The Epitect Bisulfite Kit (Qiagen) was used to convert unmethylated cytosines in genomic DNA to uracil. Touchdown PCR was used and purified PCR products were subjected to Sanger sequencing (Eton Bioscience).

**TCGA RNA-seq and DNA methylation data for validation**. Publicly available normalized beta values generated using Illumina Infinium HumanMethylation450 (HM450K) BeadChip, normalized gene expression data obtained by RNA sequencing, and clinical data were downloaded from the Broad TCGA GDAC (http://gdac.broadinstitute.org). Similarly, within TCGA data set, gene expression

was correlated with the normalized beta value at each HM450K probe using Spearman's rank correlation followed by FDR correction using Benjamini-Hochberg method. FDR *q* value of less than 0.05 was considered significant and kernel density estimation was used to plot the distribution of significant probes around the TSS ± 5 kb. Permutation tests were also performed 1000 times to obtain null distributions.

**Unsupervised hierarchical clustering**. FDR adjusted *P* values were used to determine regions or probes with the most significant methylation-expression correlation for a given gene. Unsupervised hierarchical clustering analysis based on DNA methylation status was then performed using Euclidean distance metric and Ward's linkage rule.

**Chromatin immunoprecipitation sequencing (ChIP-seq)**. Two HPVOPSCC were used for the preparation of the first generation PDX models using xeno-grafting procedures described elsewhere[17,51,52]. ChIP-seq data were analyzed with the pipeline tool Omics Pipe[53] using the ChIPseq_HOMER pipeline running HOMER v4.8[54]. The utility program homerTools was used to trim the adapters off of the raw reads prior to aligning to the reference human genome (hg19) with Bowtie v1.0.1. HOMER v4.8 was used to identify regions of the genome where more reads are present than random with default parameters for histone marks. Genomic Regions Enrichment of Annotations Tool (GREAT) prediction version 3.0.0 was performed using default settings[55]. Of note, limited data from these experiments have been previously reported in a separate analysis[17].

**Gene ontology and pathway analyses**. Single-sample gene set enrichment analysis (ssGSEA)[56] was carried out using C2, C3, and H libraries in MsigDB v5.1 (http://software.broadinstitute.org/gsea/msigdb). Gene sets of interest were expanded to show genes leading to the enrichment of certain pathways.

**Whole-exome sequencing (WES) analyses**. WES pipeline was performed on 94 samples, which included 47 patients with HPVOPSCC and 47 paired-normal samples. The genomes were realigned to the human 1000 genomes v37[57]. To generate sequence alignment and variant calls on these WES samples, we implemented our WES analysis pipeline on the cfncluster v1.3.1 of Amazon Web Service (https://github.com/awslabs/cfncluster). Short reads were mapped to the human 1000 genomes v37 by BWA-mem v0.7.12[58]. Subsequent processing was carried out with SAMtools v.1.1[59], Picard Tools v.1.96, Genome Analysis Toolkit (GATK) v2.4-9[60]. MuTect[61] version 1.1.5 and VarDict[62] was used to identify somatic mutations, indels, complex and structural variants, and germline variants by directly comparing normal and tumor data at every position of sufficient coverage. To functionally annotate genetic variants identified, we applied ANNOVAR 2017Jun01[63]. To further evaluate and filter our variants, a filter was created in which insertions, deletions, and nonsynonymous variants with ExAC and 1000 Genomes population allele frequency < 0.05 passed.

**Cell culture and siRNA knockdown**. HPV-positive cell line UM-SCC-47 was obtained from the Gutkind Laboratory at the University of California San Diego[64,65] and cultured in Dulbecco's modified Eagle's medium (Sigma Aldrich), supplemented with 10% fetal bovine serum plus penicillin (50 U ml$^{-1}$) and streptomycin (50 μg ml$^{-1}$). Cells have been authenticated by short tandem repeat (STR) profiling and tested by PCR for mycoplasma contamination. Cells were transfected with ON-TARGETplus SMART-pool siRNA for *EP300* or *MYC* (GE Dharmacon) using Lipofectamine RNAiMAX (Thermo Fisher Scientific) according to the manufacturer's protocols. A scrambled ON-TARGETplus Non-targeting pool siRNA was used as a negative control. Cell growth was assessed on the day of transfection, and at 24, 48 and 72 h after transfection using AquaBluer Solution (MultiTarget Pharmaceuticals LLC).

**Western blot**. Protein analysis was performed on lysates of UM-SCC-47 cells collected at 72 h after siRNA transfection. Primary antibodies used were c-Myc (D3N8F, 1:1000, Cell Signaling Technology), Cdk2 (M2 sc-163, 1:500, Santa Cruz Biotechnology), and GAPDH (14C10, 1:1000, Cell Signaling Technology). Immobilon Western Chemiluminescent HRP Substrate (Thermo Fishcer Scientific) reagent was used for western blot development.

**Quantitative PCR (qPCR)**. RNA was isolated from UM-SCC-47 cell pellets at 48 h after siRNA transfection using RNeasy Plus Mini Kit (Qiagen). Reverse transcription was performed using High Capacity cDNA Reverse Transcription Kit (Thermo Fisher Scientific). Quantitative reverse transcription PCR (RT-qPCR) was then performed using primers for *MYC*, *EP300*, *CDK2*, *ZNF470*, *ZNF568*, *ZNF569*, and *ACTB* obtained from TaqMan Gene Expression Assays (Thermo Fisher Scientific).

ChIP were performed using digested chromatin from UM-SCC-47 cells and the ChIP-grade histone H3K27ac (D5E4, 1:100) and c-Myc (D3N8F, 1:50) monoclonal antibodies (Cell Signaling Technology), according to the manufacturer's protocol of the SimpleChIP Enzymatic Chromatin IP Kit (Cell Signaling Technology). Purified DNA was analyzed by quantitative PCR (ChIP-qPCR) using primers for

*MYC*[20], *CDK2*, *GCK* as a negative control, *CDK4* and *NPM1* as positive controls[66]. Primers for *CDK2* and *GCK* were designed using ChIP-Atlas (http://chip-atlas.org). The sequences of primers were listed in Supplementary Table 1 and 2.

**Quantitative methylation-specific PCR (qMSP)**. DNA methylation analysis using qPCR was performed. In brief, DNA was isolated from UM-SCC-47 cell pellets at 48 h after siRNA transfection using QIAamp DNA Mini Kit (Qiagen). Epitect Bisulfite Kit was used for bisulfite conversion. TaqMan primers and probes were designed to specifically amplify the bisulfite-converted DNA for genes of interest. The sequences of TaqMan primers and probes were listed in Supplementary Table 3.

**Reporting Summary**. Further information on research design is available in the Nature Research Reporting Summary linked to this article.

## Data availability
The data sets generated from RNA-seq, MBD-seq, and ChIP-seq have been deposited in the Gene Expression Omnibus (GEO) repository with the following accession numbers: GSE112026 (RNA-seq), GSE112023 (MBD-seq), and GSE112021 (ChIP-seq). MuTect and VarDict analyses are included in Supplementary Data 5. Publicly available TCGA data were downloaded from the Broad TCGA GDAC (http://gdac.broadinstitute.org; HNSC, COAD, and BRCA). All other remaining data supporting the findings of this study are available within the Article and Supplementary Files, or available from the authors upon reasonable request. A reporting summary for this article is available as a Supplementary Information file. The source data underlying Fig. 6d–g, Supplementary Figures 6b and 6c are provided as a Source Data file.

## Code availability
Custom R script for MBD-seq analyses is available at GitHub (https://github.com/favorov/differential.coverage). The authors declare that all the other scripts generating the figures and supporting the findings of this study are available from the corresponding author upon reasonable request.

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

## Acknowledgements

We thank Bing Ren, and Naoki Nariai for insightful comments and discussions. Without their encouragement, this paper would not have materialized. The authors also thank Sayed Sadat for technical assistance. This work was supported by JSPS KAKENHI 15KK0334 (M. Ando), NIH UL1TR001442 of CTSA (K. Fisch), NIH R01DE023347 (J. Califano), and NIH P50DE019032 (J. Califano).

## Author contributions

M.A., K.M. and J.A.C. conceived and designed the experiments, developed methodology, and analyzed data. G.X., N.B., K.M.-E., M.P., K.F., A.V.F., S.Y., E.J.F., P.T. and J.A.C. developed methodology and analyzed data. M.A., Y.S., S.R., J.P., A.M., D.G., T.G., P.H., performed experiments and analyzed data. A.S., T.F., C.L., S.H., T.Y., T.I. and J.A.C. contributed materials and analyzed data. All authors contributed to the written manuscript.

## Additional information

**Competing interests:** The authors declare no competing interests.

