## [Peer Review File · Nature Communications]

Reviewers' comments:

Reviewer #1 (Remarks to the Author):

The authors investigate DNA methylation and H3K9me3 patterns in human papillomavirus-related oropharyngeal squamous cell carcinoma samples (HPVOPSCC) and integrate their results with reported RNA-seq and annotated genomic landmarks. By comparing these samples to normal mucosal tissue and cross-validating with other cancer models obtained from TCGA, they observe an HPVOPSCC tumor subtype with hypermethylation at certain TSS. These do not necessarily have high CpG content but do contain Myc binding sites and lowered expression of the associated gene.

The work performed is robust and highlights interesting key points, such as the shift of peak methylation from upstream to TSS in normal to tumor samples. This could possibly indicate a mechanistic change, which is not addressed experimentally, however. To a large part the results are also confirmatory as indicated by the authors in the Discussion. Given the lack of mechanistic insight driving the observed epigenetic changes the work remains mostly at the descriptive level. Hence, I doubt that the information contained in this manuscript is of sufficient novelty and experimental originality to merit a publication in Nature Communications.

Some additional points to consider:

1. It is not wholly clear what set of genomic ranges was used for regions excluding CGIs surrounding the TSS, first introduced in P.9 L.2. Do they constitute non-CGI genes with the additional restriction of not having CGIs in the entire window instead of -1.5kb to +0.5kb?
2. It is not stated from where the annotation tracks for CGIs stem from, or if a cutoff was used for size and score of CGIs.
3. P.10 L10. Bisulfite sequencing is used to confirm methylation status within gene-bodies. However, this is not elaborated on with whole genome MBD-Seq or substantiated with phenotypic evidence.
4. Note number of genes per subset in plots 1a, b, 2a, 4a, c, 5a, b, c, e, f, g and similar supplementary plots.

5. Try to avoid the use of red and green in the same plot (esp. 4c).

6. Was the randomly selected control set for P.13 L.6 drawn from all genes or only those that were detected in RNA-Seq? The 407 significant genes are at least constrained by detection in RNA-Seq data with expression and repressive marks being dependent, whereas the truly random sample is not.

7. Methylation profiles for ACC are indeed very different from all other ones shown (HPV-HNSCC, COAD, BRCA and HPVOPSCC). However, interpretation (P.15 L.9) has to be made with caution, when not comparing MBD sequencing quality and overall methylation status within the different cell lines.

8. Note statistical test and sample size for 6b.

9. The MYC pathway is functionally implicated via ssGSEA motif enrichment and increase of CEBBP and EP300 loss in Tumor-H. As a support for these claims, MYC levels should be determined not only by RNA-Seq, but also by direct protein evidence such as Western Blots. To link MYC to differential expression and possibly DNA methylation, changes in occupancy of MYC at select or all relevant genes needs to be determined via ChIP-qPCR or -Seq.

10. Clinical relevance should be affirmed by analyzing survival of patients with the different subtypes.

11. The authors often mention CpG Shore methylation on the basis of the observed profiles (Fig1, discussed on P.19 L.12). However, they never relate this methylation with the real position of the CGIs (i.e. meta-analyzes). To my knowledge, these sites are not rigidly positioned within a certain window around the TSS, hence, the signals could still partly be derived from CGIs. Additionally, if a promoter is considered CGI-less, that just means that it does not follow any of the current definitions. The CG content may still be high, or the CGI may be smaller than what the standard algorithm allows. However, this is never considered or discussed.

Reviewer #2 (Remarks to the Author):

Using a small cohort of HPV-related OPSCC as a model, Ando et al examined active and repressive chromatin marks, whole genome methylation and transcriptional alterations to define the relationship between transcriptional modulation and spatial changes in chromatin structure with histone and DNA methylation. The principle concept is original and the manuscript is well written. Highlight of the presented data is the identification of a common epigenetic mechanism of transcriptional dysregulation in solid tumors, in particular HPV-related OPSCC, based on a broad enrichment of repressive chromatin marks and aberrant DNA hypermethylation at transcriptional start sites independent of CpG island presence, in association with MYC network activation. However, weak points in the study design diminish its impact and the last part on MYC network activation remains largely descriptive. Additional experimental data are required to improve the added value and quality of the manuscript to be published in Nature Communications.

Major points

1. To determine functionally relevant methylation events in HPV-related OPSCC, MBD-seq and RNA-seq analysis on 47 primary tumors and 25 normal oropharyngeal mucosal tissues were conducted. The fact that primary tumors and normal tissues were obtained from different individuals and most were not matched for gender, age and other risk factors, such as smoking or alcohol, is a critical issue of the study design.
2. In Fig. 3 three methylation subtypes for the discovery cohort were identified and confirmed for selected cases of TCGA-HNC by unsupervised hierarchical clustering. Do these subtypes also exist in HPV-negative HNSCC of the TCGA-HNC cohort or in breast cancer, are they related to previously identified subgroups based on transcriptome analysis, and what is the clinical relevance of these subtypes concerning clinical and histopathological features as well as prognosis? Does the selected gene set also include HPV-related candidate genes, which were identified in previous studies, e.g. using 450K arrays?
3. PDX models derived from high and low methylation subtypes defined in the discovery cohort and cancer unaffected normal mucosa were used to demonstrate that DNA hypermethylation occurs at the TSS specifically in the highly methylated tumor. How did the authors generate a PDX with samples from unaffected normal mucosa? How many PDX models derived from tumor tissue were generated and analyzed? This question is of particular interest as recent publications report a rather low efficacy for the generation of PDX from HPV-related OPSCC with a certain risk for an in vivo selection bias. In addition, a mouse-specific selection and clonal expansion of cancer cell subclones in PDX models have been demonstrated. Did the authors observe differences in global patterns of DNA methylation, transcription alterations and chromatin marks between samples from matched PDX models and primary tumors? Representative data for the gene set shown in Fig. 3 should be provided.
4. To validate new findings from the discovery cohort and to exclude a potential methylation platform bias the authors analyzed 54 OPSCC and 20 normal samples of the TCGA-HNC cohort. Were

the normal samples selected for localization at the oropharyngeal subsite and matched for gender, age and other risk factors?

5. Similar to findings in HPV-related OPSCC, a significant increase in negative methylation-expression correlation at the TSS was also unraveled in HPV-negative HNSCC and breast cancer using CGI genes. Was this correlation also evident for nonCGI candidate genes?

6. To identify key networks associated with DNA methylation subtypes ssGSEA with overexpressed genes in highly as compared to low methylated tumors was performed. What was the rationale to restrict the analysis to the set of overexpressed candidate genes and exclude differentially expressed genes with reduced expression? ssGSEA analysis unraveled an enrichment for motifs related to MYC. In addition, reduced expression of KDM6A, a key player in epigenetic regulation, and a significant association of CEBBP/EP300 mutations was found for the hypermethylated phenotype. To further support their findings, authors should perform IHC or IF staining on tissue sections of the discovery cohort to demonstrate inverse expression of MYC and KDM6A in individual cancer cells as well as differences in their expression dependent on CEBBP/EP300 mutations. Did the authors investigate whether KDM6A expression or presence of CEBBP/EP300 mutations are associated with clinical and histopathological features or exhibits any prognostic value in the discovery cohort or TCGA-HNC?

Minor points

1. Page 7 line 1-2: the statement 'HPVOPSCC is not associated with tobacco and alcohol use' is misleading. It is true that HPV-related OPSCC are less common associated with tobacco and alcohol abuse as compared to HPV-negative HNSCC, but a substantial number of patients, including the discovery and TCGA-HNC cohorts (Fig. 3), are current or former smokers and/or drinkers.

Reviewer #1 (Remarks to the Author):

The authors investigate DNA methylation and H3K9me3 patterns in human papillomavirus-related oropharyngeal squamous cell carcinoma samples (HPVOPSCC) and integrate their results with reported RNA-seq and annotated genomic landmarks. By comparing these samples to normal mucosal tissue and cross-validating with other cancer models obtained from TCGA, they observe an HPVOPSCC tumor subtype with hypermethylation at certain TSS. These do not necessarily have high CpG content but do contain Myc binding sites and lowered expression of the associated gene.

The work performed is robust and highlights interesting key points, such as the shift of peak methylation from upstream to TSS in normal to tumor samples. This could possibly indicate a mechanistic change, which is not addressed experimentally, however. To a large part the results are also confirmatory as indicated by the authors in the Discussion. Given the lack of mechanistic insight driving the observed epigenetic changes the work remains mostly at the descriptive level. Hence, I doubt that the information contained in this manuscript is of sufficient novelty and experimental originality to merit a publication in Nature Communications.

> Thank you for your time and insightful comments. To address the mechanistic insight driving the observed epigenetic changes, we used UM-SCC-47 cell line (*CREBBP* Q1092X mutant, EP300 wild type) as a model. Knockdown of MYC suppressed growth of these cells harboring a inactivating mutation of *CREBBP* as previously reported (SFig. 6b). Firstly, we demonstrated that EP300 knockdown significantly reduced MYC expression level leading to inhibition of the proliferation of these cells, and confirmed that the EP300-mediated Myc suppression was due to reduced H3K27ac histone modification (Fig. 6d and e). We also examined the impact of *EP300* knockdown on one of the MYC target genes *CDK2*, and finally evaluated alterations in DNA methylation induced by *EP300* knockdown (Fig. 6f and g). Taken together, we think we have demonstrated the dependence of DNA methylation status on MYC pathway alteration in the *CREBBP*-deficient context, which we found in highly methylated subtype. We have extensively revised the Results and Discussion sections in accordance with your comments and suggestions. All changes in the revised manuscript have been highlighted in red.

Some additional points to consider:

1. It is not wholly clear what set of genomic ranges was used for regions excluding CGIs surrounding the TSS, first introduced in P.9 L.2. Do they constitute non-CGI genes with the additional restriction of not having CGIs in the entire window instead of -1.5kb to +0.5kb?

> By definition in this article, CGI genes have the CpG island-containing promoter (-1.5 to +0.5 kb), though they may have additional CpG island(s) outside of the promoter region. DNA methylation status at 100 bp segment with such CpG island(s) will be plotted in Fig. 1a (CGI genes, within CpG island), so we can see plots beyond -1.5kb or +0.5 kb. On the other hand, CGI genes also have various DNA methylation profiles at CpG sites outside of the island(s), i.e. shore(s) etc., and such 100bp segment will be plotted in Fig. 1b (CGI genes, outside CpG island). Thus non-overlapping 100 bp segments of an identical set of CGI genes are plotted in Fig. 1a and 1b. We have added an explanation to the Figure Legend (P.41 L.770).

2. It is not stated from where the annotation tracks for CGIs stem from, or if a cutoff was used for size and score of CGIs.

> CpG islands data was downloaded from the UCSC genome annotation database for GRCh37/hg19. We have added this in the Methods section (P.27 L.493).

3. P.10 L10. Bisulfite sequencing is used to confirm methylation status within gene-bodies. However, this is not elaborated on with whole genome MBD-Seq or substantiated with phenotypic evidence.

> According to reviewer's suggestion, we have added the corresponding MBD-seq and RNA-seq data in Fig. 2d to substantiate our findings.

4. Note number of genes per subset in plots 1a, b, 2a, 4a, c, 5a, b, c, e, f, g and similar supplementary plots.

> We have added the number of genes in Figure Legends (Fig. 1a, 1b, 2a, 4a, 4c, and 5a-g, SFig. 2, 3, and 5). Although we used protein-coding genes (n=20013) based on RefSeq, please note the different total number of genes in each cohort; 19866, 19004, and 19239 for the discovery HPVOPSCC, the TCGA validation, and the ACC cohorts, respectively. This is because we needed to merge results obtained from different datasets, i.e., DNA methylation analysis (MBD-seq or HM450K microarray) and RNA-seq.

5. Try to avoid the use of red and green in the same plot (esp. 4c).

> We have changed color in Fig. 4c. We have also replaced plots (Fig. 3b, 3c, 6a-c, and SFig. 6a) using green checkered pattern.

6. Was the randomly selected control set for P.13 L.6 drawn from all genes or only those that were detected in RNA-Seq? The 407 significant genes are at least constrained by detection in RNA-Seq data with expression and repressive marks being dependent, whereas the truly random sample is not.

> Genes (n=407) in the control set were randomly selected from the RefSeq genes (n=20013), which is not constrained by RNA-Seq data nor ChIP-seq. We generated 1000 sets of random 407 control genes to show the background signal in Fig. 4d. We have added an explanation to the Results section (P.13 L.229) and Figure Legends (P.43 L.814).

7. Methylation profiles for ACC are indeed very different from all other ones shown (HPV-HNSCC, COAD, BRCA and HPVOPSCC). However, interpretation (P.15 L.9) has to be made with caution, when not comparing MBD sequencing quality and overall methylation status within the different cell lines.

> In ACC, we did not find a shift in distribution despite the fact that the promoter region tended to be more highly associated in normal tissues as expected. We have mentioned the difference in overall methylation status and the quality issue (P.15 L.269).

8. Note statistical test and sample size for 6b.

> We have added the information accordingly in Figure Legends.

9. The MYC pathway is functionally implicated via ssGSEA motif enrichment and increase of CREBBP and EP300 loss in Tumor-H. As a support for these claims, MYC levels should be determined not only by RNA-Seq, but also by direct protein evidence such as Western Blots. To link MYC to differential expression and possibly DNA methylation, changes in occupancy of MYC at select or all relevant genes needs to be determined via ChIP-qPCR or -Seq.

> We used UM-SCC-47 cell line as a model for highly methylated phenotype (Tumor-H) , as we don't have sufficient primary tumor tissues for Western blot and CHIP analyses. To link MYC to differential expression and DNA methylation, MYC mRNA and protein levels, changes in occupancy of MYC at the target gene CDK2, and the effect on DNA methylation status were determined via the indicated methods (P.17 L.302, Fig. 6d-g).

10. Clinical relevance should be affirmed by analyzing survival of patients with the different subtypes.

> Our discovery cohort consisted of patients with stage 3/4 HPVOPSCC (AJCC/UICC 7th edition) and almost all received postoperative radiotherapy with or without concurrent chemotherapy. In consequence, only three patient, one from each subtype, experienced disease recurrence. Although the identified subtypes of HPVOPSCC cannot be used for stratification of patients treated with intensive multimodal therapy, we believe that our findings may provide clue to future de-escalation therapy. We added the relevant comment accordingly to the Discussion section (P.23 L.433).

11. The authors often mention CpG Shore methylation on the basis of the observed profiles (Fig1, discussed on P.19 L.12). However, they never relate this methylation with the real position of the CGIs (i.e. meta-analyzes). To my knowledge, these sites are not rigidly positioned within a certain window around the TSS, hence, the signals could still partly be derived from CGIs. Additionally, if a promoter is considered CGI-less, that just means that it does not follow any of the current definitions. The CG content may still be high, or the CGI may be smaller than what the standard algorithm allows. However, this is never considered or discussed.

> We agree with reviewer one that CGIs are not rigidly positioned around a window around the TSS, and that signals could still partly be derived from CGIs outside of consensus promoter regions. However, we chose the most commonly reported and accepted consensus definitions to allow for our results to be compared to prior studies. We agree that if a promoter is CGI-less, that it doesn't necessarily conform to current definitions. The implications of our data are indeed that the definition of a promoter regions that responds to epigenetic alteration may not universally apply, and that CGI that don't fit current definitions may still be responsive to methylation and histone alteration. We have included this in comments in the discussion (P.20 L.358).

Reviewer #2 (Remarks to the Author):

Using a small cohort of HPV-related OPSCC as a model, Ando et al examined active and repressive chromatin marks, whole genome methylation and transcriptional alterations to define the relationship between transcriptional modulation and spatial changes in chromatin structure with histone and DNA methylation. The principle concept is original and the manuscript is well written. Highlight of the presented data is the identification of a common epigenetic mechanism of transcriptional dysregulation in solid tumors, in particular HPV-related OPSCC, based on a broad enrichment of repressive chromatin marks and aberrant DNA hypermethylation at transcriptional start sites independent of CpG island presence, in association with MYC network activation. However, weak points in the study design diminish its impact and the last part on MYC network activation remains largely descriptive. Additional experimental data are required to improve the added value and quality of the manuscript to be published in Nature Communications.

> Thank you for your time and constructive comments. To address the mechanistic insight on MYC network and DNA methylation, we have performed additional experiments using UM-SCC-47 cell line (*CREBBP* Q1092X mutant, *EP300* wild type) as a model for highly methylated subtype. We have extensively revised the Results and Discussion sections in accordance with your comments and suggestions. All changes in the revised manuscript have been highlighted in red.

Major points

1. To determine functionally relevant methylation events in HPV-related OPSCC, MBD-seq and RNA-seq analysis on 47 primary tumors and 25 normal oropharyngeal mucosal tissues were conducted. The fact that primary tumors and normal tissues were obtained from different individuals and most were not matched for gender, age and other risk factors, such as smoking or alcohol, is a critical issue of the study design.

> Thank you for your insightful comments. It is challenging to obtain adequate normal tissues from matched healthy individuals. In addition, tumor adjacent tissues of the cancer-affected individuals are known to carry genetic and epigenetic alterations that can introduce significant bias. We have added this issue as one of the limitations of our study (P.24 L.438).

2. In Fig. 3 three methylation subtypes for the discovery cohort were identified and confirmed for selected

cases of TCGA-HNC by unsupervised hierarchical clustering. Do these subtypes also exist in HPV-negative HNSCC of the TCGA-HNC cohort or in breast cancer, are they related to previously identified subgroups based on transcriptome analysis, and what is the clinical relevance of these subtypes concerning clinical and histopathological features as well as prognosis? Does the selected gene set also include HPV-related candidate genes, which were identified in previous studies, e.g. using 450K arrays?

> We couldn't clearly identify DNA methylation subtypes among TCGA HPV-negative HNSCC nor breast cancer when using the selected 59 genes (Fig. 3a) determined by MBD-seq for our discovery HPVOPSCC cohort. We also tried to determine an optimized gene set using each TCGA cohort, but probably due to limited coverage provided by HM450K microarray, subsequently failed to identify DNA methylation subtypes. Although we think that integrated DNA methylation and gene expression analysis might define subtypes in these tumor types, the specific gene set according to tumor type is needed for successful clustering and the discovery analysis would preferably be broader than using HM450K microarray. We have added the statement in the Discussion section accordingly (P.21 L.394).

3. PDX models derived from high and low methylation subtypes defined in the discovery cohort and cancer unaffected normal mucosa were used to demonstrate that DNA hypermethylation occurs at the TSS specifically in the highly methylated tumor. How did the authors generate a PDX with samples from unaffected normal mucosa? How many PDX models derived from tumor tissue were generated and analyzed? This question is of particular interest as recent publications report a rather low efficacy for the generation of PDX from HPV-related OPSCC with a certain risk for an in vivo selection bias. In addition, a mouse-specific selection and clonal expansion of cancer cell subclones in PDX models have been demonstrated. Did the authors observe differences in global patterns of DNA methylation, transcription alterations and chromatin marks between samples from matched PDX models and primary tumors? Representative data for the gene set shown in Fig. 3 should be provided.

> We generated two tumor PDX models and compared the RNA-seq gene expression profile to the profile for the corresponding parental tissue. Pearson correlation coefficients were 0.83 for PDX1 (Tumor-L) and 0.9 for PDX2 (Tumor-H), and both P values were below 10^{-16} (ref. 17). We have mentioned this confirmatory study in the Results section (P.12 L.200) We have also provided visualization of DNA methylation comparing PDXs and parent tumors for representative genes shown in Fig. 3a (SFig. 4).

> We didn't generate PDX models from normal mucosa. The statement in the submitted manuscript was misleading and corrected accordingly (P.11 L.196). Chromatin marks were

investigated using normal tissues and tumor PDX models by ChIP-seq, thus data for parental samples cannot be obtained.

4. To validate new findings from the discovery cohort and to exclude a potential methylation platform bias the authors analyzed 54 OPSCC and 20 normal samples of the TCGA-HNC cohort. Were the normal samples selected for localization at the oropharyngeal subsite and matched for gender, age and other risk factors?

> The normal samples of the TCGA-HNC cohort are not necessarily matched oropharyngeal tissue. As the reviewer pointed out, there are several limitations in establishing adequate controls using TCGA cohort. Nonetheless, TCGA is still the best database available to date, which can be used for validation purposes. We have added this issue as one of the limitations of our study (P.24 L.438).

5. Similar to findings in HPV-related OPSCC, a significant increase in negative methylation-expression correlation at the TSS was also unraveled in HPV-negative HNSCC and breast cancer using CGI genes. Was this correlation also evident for nonCGI candidate genes?

> We didn't find significant increase in negative methylation-expression correlation at the TSS for nonCGI genes in HPV-negative HNSCC and breast cancer as in HPV-related OPSCC.

6. To identify key networks associated with DNA methylation subtypes ssGSEA with overexpressed genes in highly as compared to low methylated tumors was performed. What was the rationale to restrict the analysis to the set of overexpressed candidate genes and exclude differentially expressed genes with reduced expression? ssGSEA analysis unraveled an enrichment for motifs related to MYC. In addition, reduced expression of KDM6A, a key player in epigenetic regulation, and a significant association of CREBBP/EP300 mutations was found for the hypermethylated phenotype. To further support their findings, authors should perform IHC or IF staining on tissue sections of the discovery cohort to demonstrate inverse expression of MYC and KDM6A in individual cancer cells as well as differences in their expression dependent on CREBBP/EP300 mutations. Did the authors investigate whether KDM6A expression or presence of CREBBP/EP300 mutations are associated with clinical and histopathological features or exhibits any prognostic value in the discovery cohort or TCGA-HNC?

> We apologize that the word "overexpressed" was a mistake as we used all RefSeq genes in RNA-seq gene expression. We have corrected the sentence in the revised manuscript (P.16 L.279).

> Regarding MYC expression and *CREBBP/EP300* mutations, we used UM-SCC-47 cell line (*CREBBP* Q1092X mutant, *EP300* wild type) as an experimental model for highly methylated phenotype, and demonstrated the dependence of DNA methylation status on MYC pathway alteration in the *CREBBP*-deficient context (Fig. 6d-g). Because our discovery cohort consisted of patients with stage 3/4 HPVOPSCC (AJCC/UICC 7th edition), almost all received postoperative radiotherapy with or without concurrent chemotherapy. In consequence, only three patient, one from each subtype, experienced disease recurrence and we didn't find any prognostic value in the discovery cohort. We have added this issue as one of the limitations of our study (P.23 L.433).

> As for reduced expression of *KDM6A*, we have decided to delete the sentence in the Results section because we couldn't provide further supportive information.

Deleted: "A key gene involved in epigenetic regulation with reduced expression in the hypermethylated phenotype is the tumor suppressor *KDM6A*, which specifically targets H3K27me3 repressive marks for demethylation"

Minor points

1. Page 7 line 1-2: the statement 'HPVOPSCC is not associated with tobacco and alcohol use' in misleading. It is true that HPV-related OPSCC are less common associated with tobacco and alcohol abuse as compared to HPV-negative HNSCC, but a substantial number of patients, including the discovery and TCGA-HNC cohorts (Fig. 3), are current or former smokers and/or drinkers.

> We have accordingly corrected the statement as follows (P.7 L.103).

"Unlike traditional head and neck squamous cell carcinoma (HNSCC), the major risk factors for HPVOPSCC are not tobacco or alcohol use, and less common somatic mutations in key cancer genes implies that epigenetic mechanisms might drive oncogenesis."

Thank you for the time and effort in reviewing our manuscript. We appreciate your comments, and believe they have helped us improve our work.

Reviewers' comments:

Reviewer #1 (Remarks to the Author):

With this revision the authors addressed and improved numerous issues. Missing information regarding experimental conditions, points 1., 2., 3., 4., 6., 8., was added. Additionally, figure appearance changes were also made, satisfying 5. and qualifiers to indicate the extent of statements were made for 7.

One major concern was regarding mechanistic details of the putative MYC-Hypermethylation axis. The authors attempted to address this using the UM-SCC-47 cell line mutated for CREBBP, a histone acetyl transferase closely related to EP300. This mimics a proportion of the background of analyzed hyper-methylated tumor samples, since 4 of 13, harbor a mutation in CREBBP. It would be important to know if the mutations occur primarily in the HAT domain as has been reported for other cancer types [1]. The authors replicate previous findings by showing that MYC expression and cell proliferation of these cells, is dependent on the presence of EP300, possibly due to a decrease in K27ac at the MYC locus (Fig. 6e). As a consequence, MYC occupancy at CDK2 and known MYC targets CDK4 as well as NPM1 drops markedly (Fig 6f, gene names need to be italicized). Furthermore, a reduction in DNA methylation can be observed in three genes (Fig. 6g).

In the hyper-methylated samples MYC is not differentially expressed compared to samples with lower-methylation or potentially control (Fig. 6b). MYC was mainly implicated through motif enrichment in differentially expressed genes (Fig. 6a, S6a). In the opinion of this reviewer a loss of MYC expression and subsequent hypo-methylation in the cell line does not necessarily substantiate any insights which would lead to a further elucidation of the hyper-methylation found in tumors with consistent MYC expression.

9. Because of sample availability, the authors probe MYC protein levels and MYC occupancy at three genes in the UM-SCC-47 cell line instead. These were chosen as representatives of a gene set with a negative expression correlation with that of MYC. As a control at least one gene should be used that does not show any expression correlation with MYC. This will help clarify the extent of DNA methylation loss upon EP300 knockdown.

10. Could not be resolved experimentally due to a too small sample size, as an alternative, the added paragraph does explain this limitation. However, the added paragraph also highlights the use of unmatched normal oropharyngeal tissues to compare to the tumor samples. With a large part of the study focusing on epigenetic differences, this can be an issue.

11. The addition of the comments in the discussion is fitting but does not conclusively answer this point. To elaborate the question: with the way the data is presented no conclusions can be drawn on how CGI-shore methylation is related to the CGI they border on. Using meta-analyses of CGIs and their shores, methylation density can be related to, for example, the distance from the CGI.

12. A point not raised earlier, L.299 describes the enrichment of MYCMAX_01 motifs as significant. It should be stated at what level it is significant and if the p-value (0.00132) or the multiple testing corrected p-value (FDR, 0.46) was used. With regards to MYC-TARGETS_V1, even non-significant p-values close to the chosen significance threshold do not indicate a trend and should not be labeled as such.

13. Were the expression quantiles (Q1 to Q4) used in Figure 1 based on their expression in the tumor condition, the normal condition or a separate condition for each the tumor and normal graphs?

[1] Inthal, A., Zeitlhofer, P., Zeginigg, M., Morak, M., Grausenburger, R., Fronkova, E., ... Panzer-Grümayer, R. (2012). CREBBP HAT domain mutations prevail in relapse cases of high hyperdiploid

childhood acute lymphoblastic leukemia. *Leukemia*, 26(8), 1797–1803.
<http://doi.org/10.1038/leu.2012.60>

Reviewer #2 (Remarks to the Author):

The authors adequately addressed most questions and concerns, which were raised by both reviewers, and included new experimental data in the revised manuscript using the UM-SCC-47 cell line as a model to provide mechanistic insight on the MYC network and its role in DNA methylation. However, the authors should provide some additional data to substantiate the relevance of this experimental model as outlined in more detail below:

1. EP300 knockdown in UM-SCC47 cells was conducted by siRNA technology and the authors should provide data on silencing efficacy on transcript and protein levels as supplemental data.
2. Three representative genes (ZNF470, ZNF568, and ZNF569) were selected for further analysis in UM-SCC47 cells upon EP300 silencing. What was the rationale to select these candidate genes taking into account that the Pearson correlation coefficient for all three genes in the TCGA validation cohort is rather low and does not reach statistical significance (Supplementary Fig. 6d). How many and which genes of Fig. 3a exhibit a statistically significant correlation with MYC concerning transcript levels in both the discovery and validation cohorts?
3. Fig. 6g shows a significant but minor decrease in DNA methylation of ZNF470, ZNF568, and ZNF569 upon EP300 silencing in UM-SCC47 cells. Is this decrease in DNA methylation accompanied by a significant increase in transcript levels?
4. To further support their model, authors should perform IHC or IF staining on tissue sections of the discovery cohort to demonstrate high MYC expression in individual cancer cells of hypermethylated tumors as well as differences in MYC expression dependent on CREBBP mutations.

Reviewer #1 (Remarks to the Author):

With this revision the authors addressed and improved numerous issues. Missing information regarding experimental conditions, points 1., 2., 3., 4., 6., 8., was added. Additionally, figure appearance changes were also made, satisfying 5. and qualifiers to indicate the extent of statements were made for 7.

One major concern was regarding mechanistic details of the putative MYC-Hypermethylation axis. The authors attempted to address this using the UM-SCC-47 cell line mutated for CREBBP, a histone acetyl transferase closely related to EP300. This mimics a proportion of the background of analyzed hyper-methylated tumor samples, since 4 of 13, harbor a mutation in CREBBP. It would be important to know if the mutations occur primarily in the HAT domain as has been reported for other cancer types [1]. The authors replicate previous findings by showing that MYC expression and cell proliferation of these cells, is dependent on the presence of EP300, possibly due to a decrease in K27ac at the MYC locus (Fig. 6e). As a consequence, MYC occupancy at CDK2 and known MYC targets CDK4 as well as NPM1 drops markedly (Fig 6f, gene names need to be italicized). Furthermore, a reduction in

DNA methylation can be observed in three genes (Fig. 6g).

In the hyper-methylated samples MYC is not differentially expressed compared to samples with lower-methylation or potentially control (Fig. 6b). MYC was mainly implicated through motif enrichment in differentially expressed genes (Fig. 6a, S6a). In the opinion of this reviewer a loss of MYC expression and subsequent hypo-methylation in the cell line does not necessarily substantiate any insights which would lead to a further elucidation of the hyper-methylation found in tumors with consistent MYC expression.

> We appreciated reviewer#1's insightful comments. We found total five CREBBP mutations in our discovery cohort; E1566X in Tumor-L, and Q771X, R1446C, E1550K, and Q2202_Q2203del in Tumor-H, four of which affect HAT (histone acetyl transferase, location:1342-1649) domain as has been reported for other cancer types [ref 22 and 23]. The UM-SCC-47 cells also harbor truncating Q1092X mutation lacking the HAT domain. We have added the information accordingly in the Result section (P.16 L.247). We do agree with the reviewer that it can be challenging for a single cell line to be broadly reflective of primary tumor biology even if they share a common histologic type due to selection pressures on adherent cell cultures.

9. Because of sample availability, the authors probe MYC protein levels and MYC occupancy at three genes in the UM-SCC-47 cell line instead. These were chosen as representatives of a gene set with a negative expression correlation with that of MYC. As a control at least one gene should be used that does not show any expression correlation with MYC. This will help clarify the extent of DNA methylation loss upon EP300 knockdown.

> According to the reviewer's suggestion, we have added GCK gene, which didn't have any expression correlation with MYC both in the discovery and TCGA cohorts, as a control in Fig. 6f. This gene doesn't have any MYC binding site.

10. Could not be resolved experimentally due to a too small sample size, as an alternative, the added paragraph does explain this limitation. However, the added paragraph also highlights the use of unmatched normal oropharyngeal tissues to compare to the tumor samples. With a large part of the study focusing on epigenetic differences, this can be an issue.

> We agree with reviewer#1 on this issue, but it is still challenging to obtain adequate normal tissues from matched healthy individuals. In this study we used tissues from non-cancer uvulopalatopharyngoplasty controls rather than to use tumor adjacent tissues, which are known to carry genetic and epigenetic alterations.

11. The addition of the comments in the discussion is fitting but does not conclusively answer this point. To elaborate the question: with the way the data is presented no conclusions can be drawn on how CGI-shore methylation is related to the CGI they border on. Using meta-analyses of CGIs and their shores, methylation density can be related to, for example, the distance from the CGI.

> We agree with reviewer#1 and decided to avoid the word "CGI-shore." In the revised manuscript, we use "1 kb upstream of the TSS" or "CpG island promoter" instead.

12. A point not raised earlier, L.299 describes the enrichment of MYCMAX_01 motifs as significant. It should be stated at what level it is significant and if the p-value (0.00132) or the multiple testing corrected p-value (FDR, 0.46) was used. With regards to MYC-TARGETS_V1, even non-significant p-values close to the chosen significance threshold do not indicate a trend and should not be labeled as such.

> According to the reviewer's comments, we have revised our manuscript (P.16 L.252).

13. Were the expression quantiles (Q1 to Q4) used in Figure 1 based on their expression in the tumor condition, the normal condition or a separate condition for each the tumor and normal graphs?

> The quartiles (Q1 to Q4) were based on a separate condition, i.e. the expression levels in either tumor or normal samples. We have added the explanation in the Figure Legends.

Reviewer #2 (Remarks to the Author):

The authors adequately addressed most questions and concerns, which were raised by both reviewers, and included new experimental data in the revised manuscript using the UM-SCC-47 cell line as a model to provide mechanistic insight on the MYC network and its role in DNA methylation. However, the authors should provide some additional data to substantiate the relevance of this experimental model as outlined in more detail below:

1. EP300 knockdown in UM-SCC47 cells was conducted by siRNA technology and the authors should provide data on silencing efficacy on transcript and protein levels as supplemental data.

> We have added EP300 mRNA level accordingly to SFig. 6c.

2. Three representative genes (ZNF470, ZNF568, and ZNF569) were selected for further analysis in UM-SCC47 cells upon EP300 silencing. What was the rationale to select these candidate genes taking into account that the Pearson correlation coefficient for all three genes in the TCGA validation cohort is rather low and does not reach statistical significance (Supplementary Fig. 6d). How many and which genes of Fig. 3a exhibit a statistically significant correlation with MYC concerning transcript levels in both the discovery and validation cohorts?

> A total of 18 genes in Fig. 3a showed a significant ($p < 0.05$) negative correlation with MYC transcript levels in the discovery cohort, and only one of which (*IFNAR1*) reached significance ($p = 0.04$) in the TCGA cohort. Because quantitative methylation specific PCR (qMSP) primers for *IFNAR1* gene was difficult to design, we selected these three genes primarily based on the significance in the discovery cohort and in terms of primer design as well.

3. Fig. 6g shows a significant but minor decrease in DNA methylation of ZNF470, ZNF568, and ZNF569 upon EP300 silencing in UM-SCC47 cells. Is this decrease in DNA methylation accompanied by a significant increase in transcript levels?

> We have added RT-qPCR for these three genes upon EP300 knockdown and found a significant increase in mRNA expression levels except for ZNF470 (Fig. 6g, right panel).

4. To further support their model, authors should perform IHC or IF staining on tissue sections of the discovery cohort to demonstrate high MYC expression in individual cancer cells of hypermethylated tumors as well as differences in MYC expression dependent on CREBBP mutations.

> Our discovery cohort samples were collected in 2001, and have undergone multiple genomic analyses that have unfortunately exhausted these tissues.

REVIEWERS' COMMENTS:

Reviewer #1 (Remarks to the Author):

The authors provide an additional revision of their manuscript. They added the requested information and made adequate corrections. In my opinion the manuscript can now be accepted for publication.

Reviewer #2 (Remarks to the Author):

The authors adequately answered all questions and concerns of both reviewers and only minor comments remain:

1. page 6 line 64 correct "These findings are complemented by ..."
2. authors should consider confirmation of a significant association between CREBBP/EP300 mutations and the hypermethylation phenotype in the TCGA HPVOPSCC cohort.
3. page 16 line 251 correct Fig. 3c

REVIEWERS' COMMENTS:

Reviewer #1 (Remarks to the Author):

The authors provide an additional revision of their manuscript. They added the requested information and made adequate corrections. In my opinion the manuscript can now be accepted for publication.

> We would appreciate reviewer#1 for the careful and constructive review of our manuscript.

Reviewer #2 (Remarks to the Author):

The authors adequately answered all questions and concerns of both reviewers and only minor comments remain:

1. page 6 line 64 correct "These findings are complemented by ..."

> Thank you for pointing it out. We have corrected the error in the final manuscript.

2. authors should consider confirmation of a significant association between CREBBP/EP300 mutations and the hypermethylation phenotype in the TCGA HPVOPSCC cohort.

> We appreciate reviewer#2's insightful comments. According to cBioPortal (<http://www.cbioportal.org>), prevalence of CREBBP/EP300 mutations in TCGA HPVOPSCC cohort were 3 out of 15 (20.0%) and 2 out of 19 (10.5%) among hyper- and hypomethylation subtype, respectively (P = 0.63, Fisher's exact test). Given the modest cohort size of HPV related oropharynx cancers in TCGA, this limits the power to define this association.

3. page 16 line 251 correct Fig. 3c

> Thank you for pointing it out. We have corrected the error in the final manuscript.